

**PM2.5 concentrations based on near-surface visibility at 4011 sites in the Northern**
**Hemisphere from 1959 to 2022**
Hongfei Hao[1], Kaicun Wang[2], Guocan Wu[1], Jianbao Liu[2], Jing Li[3]
[1]Global Change and Earth System Science, Faculty of Geographical Science, Beijing Normal
University, Beijing 100875, China
[2]Institute of Carbon Neutrality, Sino French Institute of Earth System Science, College Urban and
Environmental Sciences, Peking University, Beijing 100871, China
[3]Institute of Carbon Neutrality, Sino French Institute of Earth System Science, Department of
Atmospheric and Oceanic Sciences, School of Physics, Peking University, Beijing 100871, China
Corresponding Author: Email: kcwang@pku.edu.cn
**Abstract**
Long-term PM2.5 data are needed to study the atmospheric environment, human health, and climate
change. PM2.5 measurements are sparsely distributed and of short duration. In this study, daily PM2.5
concentrations are estimated from 1959 to 2022 using a machine learning method at 4011 terrestrial
sites in the Northern Hemisphere based on hourly atmospheric visibility data, which are extracted
from the Meteorological Terminal Aviation Routine Weather Report (METAR). PM2.5 monitoring is
the target of machine learning, and atmospheric visibility and other related variables are the inputs.
The training results show that the slope between the estimated PM2.5 concentration and the
monitored PM2.5 concentration is $0.946 \pm 0.0002$ within the 95% confidence interval (CI), the
coefficient of determination ($R^2$) is 0.95, the root mean square error (RMSE) is 7.0 μg/m$^3$, and the
mean absolute error (MAE) is 3.1 μg/m$^3$. The test results show that the slope between the predicted
PM2.5 concentration and the monitored PM2.5 concentration is $0.862 \pm 0.0010$ within a 95% CI, the
$R^2$ is 0.80, the RMSE is 13.5 μg/m$^3$, and the MAE is 6.9 μg/m$^3$. The multiyear mean PM2.5
concentrations from 1959 to 2022 in the United States, Canada, Europe, China, and India are 11.2
μg/m$^3$, 8.2 μg/m$^3$, 20.1 μg/m$^3$, 51.3 μg/m$^3$ and 88.6 μg/m$^3$, respectively. PM2.5 is low and continues
to decrease from 1959 to 2022. PM2.5 in the United States increases slightly at a rate of 0.38
μg/m$^3$/decade from 1959 to 1990 and decreases at a rate of -1.32 μg/m$^3$/decade from 1991 to 2022.
Trends in Europe are positive (5.69 μg/m$^3$/decade) from 1959 to 1972 and negative (-1.91
μg/m$^3$/decade) from 1973 to 2022. Trends in China and India are increasing (3.04 and 3.35
μg/m$^3$/decade, respectively) from 1959 to 2012 and decreasing (-38.82 and -42.84 μg/m$^3$/decade,
respectively) from 2013 to 2022. The dataset is available at National Tibetan Plateau / Third Pole
Environment Data Center (https://doi.org/10.11888/Atmos.tpdc.301127) (Hao et al., 2024).
**Keywords**
Fine particulate matter; PM2.5; Visibility; Machine learning; Dataset.
**1 Introduction**
Fine particulate matter (PM2.5) refers to particulate matter suspended in air with an aerodynamic
diameter of less than 2.5 micrometers. PM2.5 has various shapes and is composed of complex
components, such as inorganic salts (e.g., sulfate, nitrate, and ammonium), as well as organic carbon



and elemental carbon, metallic elements, and organic compounds (Chen et al., 2020; Fan et al.,
2021). $PM_{2.5}$ can be emitted directly into the atmosphere (Viana et al., 2008; Zhang et al., 2019) and
generated through photochemical reactions and transformations (Guo et al., 2014). $PM_{2.5}$ exhibits
high concentrations near emission sources, which gradually decreases with distance. Due to the
small size and longer life span of $PM_{2.5}$ compared with coarse particulate matter, it can be
transported over long distances by atmospheric movements, leading to wide-ranging impacts.
Studies indicate that regional transport contributes significantly to local $PM_{2.5}$ (Wang et al., 2014;
Chen et al., 2020).
$PM_{2.5}$ reduces atmospheric visibility and facilitates the formation of fog and haze conditions (Fan
et al., 2021). Direct and indirect effects on solar radiation in the atmosphere (Albrecht, 1989;
Ramanathan et al., 2001; Bergstrom et al., 2007; Chen et al., 2022) alter the energy balance and the
number of condensation nuclei, thereby influencing atmospheric circulation and the water cycle
(Wang et al., 2012; Liao et al., 2015; Samset et al., 2019; Li et al., 2022).
$PM_{2.5}$ is also known as respirable particulate matter. Due to its complex composition, $PM_{2.5}$ may
carry toxic substances that can significantly impair human health. The World Health Organization
states explicitly that $PM_{2.5}$ is more harmful than coarse particles, and long-term exposure to high
$PM_{2.5}$ concentrations increases the risk of respiratory diseases, cardiovascular diseases, and lung
cancer (Lelieveld et al., 2015), regardless of a country's development status. A Global Burden of
Diseases study revealed that exposure to environmental $PM_{2.5}$ causes thousands of deaths and
millions of lung diseases annually (Chafe et al., 2014; Kim et al., 2015; Cohen et al., 2017).
$PM_{2.5}$ is an important parameter for assessing particulate matter pollution and air quality (Wang et
al., 2012). $PM_{2.5}$ can lead to soil acidification, water pollution, disruption of plant respiration, and
ecological degradation (Wu and Zhang, 2018; Liu et al., 2019). Due to globalization and economic
integration, preventing and controlling particulate matter pollution is a challenge at city, country
and global scales.
Therefore, long-term $PM_{2.5}$ data are needed for studies on the environment, human health, and
climate change. At present, ground-based measurements, chemical models, and estimations of
alternatives are the primary sources of $PM_{2.5}$ data.
Ground-based measurements are the most effective means to measure $PM_{2.5}$. $PM_{2.5}$ monitoring has
been ongoing since the 1990s in North America and Europe (Van Donkelaar et al., 2010), and large-
scale $PM_{2.5}$ monitoring has been implemented in other regions since 2000, including China in 2013
(Liu et al., 2017). As a result, the records for $PM_{2.5}$ are short, with only a few years of data available
in many countries. The scarcity of $PM_{2.5}$ measurements makes it challenging to provide long-term
historical data for research.
Reanalysis datasets provide estimates of long-term particulate matter concentrations. The Modern-
Era Retrospective Analysis for Research and Applications version 2 (MERRA-2) is a reanalysis
dataset from NASA that uses the Goddard Earth Observing System version 5 (GEOS-5), which has
provided global $PM_{2.5}$ data since 1980 (Buchard et al., 2015; Buchard et al., 2016; Buchard et al.,
2017; Gelaro et al., 2017; Sun et al., 2019). The MERRA-2 surface $PM_{2.5}$ assessment results are
more consistent between observations located in rural areas, as cities and suburban areas are affected
by high local emissions that do not represent the estimated grid average. Due to the lack of nitrate





and low organic carbon emissions in GOCART, there is a difference in the total amount of PM$_{2.5}$
during winter in the western United States, and sea salt aerosols are overestimated (Buchard et al.,
2017). Another reanalysis dataset is the Copernicus Atmosphere Monitoring Service (CAMS) global
reanalysis, which is a global reanalysis dataset of the atmospheric composition produced by the
European Centre for Medium-Range Weather Forecasts (ECMWF) and has provided PM$_{2.5}$ data
since 2003 (Che et al., 2014; Inness et al., 2019). The validation of PM$_{2.5}$ for CAMS shows severe
overestimations in some areas (Ali et al., 2022; Jin et al., 2022). Although reanalysis provides long-
term PM$_{2.5}$ data, the uncertainty in emission inventories increases the uncertainty in PM$_{2.5}$, which
remains challenging (Granier et al., 2011).
Many studies have employed statistical methods, machine learning, and deep learning methods to
estimate PM$_{2.5}$ concentrations based on aerosol optical depth (AOD). Van Donkelaar et al. (2021)
utilized satellite AOD, chemical transport models, and ground-level measurements of AOD to
estimate monthly PM$_{2.5}$ concentrations and their uncertainties over global land from 1998 to 2019,
and there are several related studies (Van Donkelaar et al., 2010; Boys et al., 2014; Van Donkelaar
et al., 2015; Van Donkelaar et al., 2016; Hammer et al., 2020). Many studies have been conducted
at the regional scale, such as in the United States (Beckerman et al., 2013), China (Wei et al., 2019b;
Xue et al., 2019; Wei et al., 2020a; He et al., 2021; Wei et al., 2021), and India (Mandal et al., 2020).
Although the PM$_{2.5}$ data derived from satellite retrievals have high spatial coverage, the temporal
range depends entirely on the satellite retrievals. The estimation of PM$_{2.5}$ based on satellite products
is also limited by bright surfaces, cloud conditions (Wei et al., 2019a) and resolution (Nagaraja Rao
et al., 1989; Hsu et al., 2017).
Another alternative for estimating PM$_{2.5}$ concentrations is the atmospheric horizontal visibility,
which is the maximum distance at which observers with normal visual acuity can discern target
contours under current weather conditions. In addition to manual observations, automated visibility
measurements were implemented early, typically relying on the aerosol scattering principle (Wang
et al., 2009; Zhang et al., 2020). Visibility and PM$_{2.5}$ are measurements of near-surface aerosols.
They describe atmospheric transparency and are used to describe atmospheric pollution. Long-term
visibility records have been used to quantify long-term aerosol properties (Molnár et al., 2008; Wang
et al., 2009; Zhang et al., 2017; Zhang et al., 2020). Visibility observation stations are densely
distributed across the country. Compared to satellite-retrieved AOD data, visibility observations
have longer historical records dating back to the early 20th century (Noaa et al., 1998; Boers et al.,
2015), are not affected by cloud interference and provide continuous measurements.
Visibility has been used as a proxy for PM$_{2.5}$ (Huang et al., 2009) and to estimate PM$_{2.5}$ (Liu et al.,
2017; Li et al., 2020; Singh et al., 2020). Singh et al. (2020) analyzed the air quality in East Africa
from 1974 to 2018 using visibility data. Liu et al. (2017) developed a statistical model and utilized
ground-level visibility data to estimate long-term PM$_{2.5}$ concentrations in China from 1957 to 1964
and 1973 to 2014. Gui et al. (2020) proposed a method to establish a virtual ground observation
network for PM$_{2.5}$ in China using extreme gradient boosting modeling in 2018. Zeng et al. (2021)
used LightGBM to establish a virtual network for hourly PM$_{2.5}$ concentrations in China in 2017.
Zhong et al. (2021; 2022) used LightGBM to predict 6-hour PM$_{2.5}$ concentrations based on visibility,
temperature, and relative humidity in China from 1960 to 2020. Meng et al. (2018) utilized a random
forest model to estimate the daily PM$_{2.5}$ components in the United States from 2005 to 2015. These
studies have provided various methods for estimating PM$_{2.5}$ using visibility data. However, some



have focused on only methodological innovations without providing long-term trends in PM$_{2.5}$.
Other studies offer long-term trends, but the primary focus was at urban and national scales. There
are few studies on long-term and high-temporal-resolution PM$_{2.5}$ at the global scale or across
different countries.
This study uses a convenient, accurate, and easily understandable machine learning approach to
estimate daily PM$_{2.5}$ concentrations based on visibility at 4,011 land-based sites from 1959 to 2022.
We also provide the long-term trends and characteristics of PM$_{2.5}$ in different regions. The PM$_{2.5}$
dataset provides support for climate change, human health, and pollution control research. First, we
build a machine learning model and then analyze the importance of the variables. Second, we
evaluate the model's performance and predictive ability. Third, we discuss the errors and limitations
of the dataset. Fourth, we compare the estimated PM$_{2.5}$ with the other datasets. Finally, we analyze
the spatial-temporal distributions of PM$_{2.5}$.
**2 Data and methods**
**2.1 Study Area**
The study area includes Canada, the United States, Europe, China, and India in the Northern
Hemisphere. The distributions of visibility stations (a) and the PM$_{2.5}$ monitoring sites (b) in each
region are shown in Figure 1. The number of visibility stations is 3177, and a total of 4011 PM$_{2.5}$
monitoring sites are selected for this study, with 1110 sites in the United States, 304 sites in Canada,
834 sites in Europe, 1557 sites in China, and 206 sites in India.

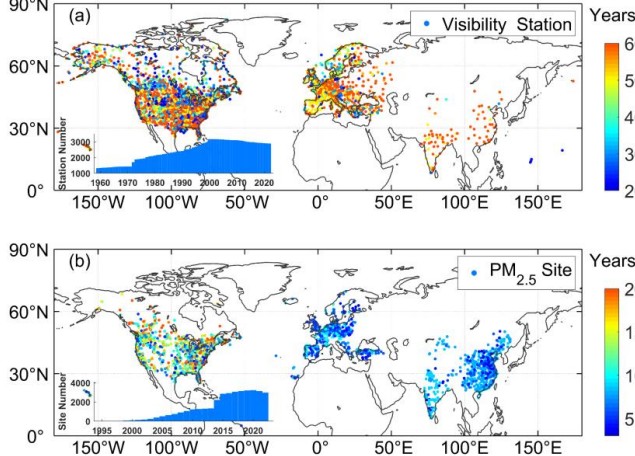


**Figure 1** Study area and the distribution of visibility stations from 1959 to 2022 (a) and PM$_{2.5}$
monitoring sites from 1995 to 2022 (b). The color of marker (circle) represents that the length of
the observation record of visibility and PM$_{2.5}$ observations. The bar chart shows the number of
visibility stations and PM$_{2.5}$ monitoring sites per year. The number of visibility stations is 3177.The
number of PM$_{2.5}$ sites is 4011 in this study (1110 in the United States, 304 in Canada, 834 in Europe,
1557 in China, and 206 in India).
**2.2 PM$_{2.5}$ Data**



### 2.2.1 PM$_{2.5}$ Data in the United States

The hourly PM$_{2.5}$ data for the United States from 1998 to 2022 are sourced from the Air Data System (AQS), which are available at https://www.epa.gov/aqs. The AQS provides PM$_{2.5}$ mass monitoring and routine chemical speciation data and contains other ambient air pollution data collected by the Environmental Protection Agency (EPA), state, local, and tribal air pollution control agencies from thousands of monitors, comprising the Federal Reference Method (FRM) and Federal Equivalent Method (FEM). The primary purpose of both methods is to assess compliance with the PM$_{2.5}$ National Ambient Air Quality Standards (NAAQS). FRMs include in-stack particulate filtration, and FEMs include beta-attenuation monitoring, very sharp cut cyclones, and tapered element oscillating microbalances (TOEMs). The measurement precision is $\pm$ (1~2) µg/m$^3$ (hour) (Gilliam and Hall, 2016). The TEOM and beta-attenuation are automatic and near real-time monitoring methods. The TEOM, which is based on gravity, measures the mass of particles collected on filters by monitoring the frequency changes in tapered elements. The beta-attenuation method uses beta-ray attenuation and particle mass to measure the PM$_{2.5}$ concentration. In this study, we use two PM$_{2.5}$ measurement methods, FRM/FEM (88101) and non-FRM/FEM (88502). The 88502 monitors are "FRM-like" but are not used for regulatory purposes. Both the 88101 and 88502 monitors are used for reporting daily Air Quality Index values.

We set the conditions that each PM$_{2.5}$ monitoring event have a minimum of 3 years and more than 1000 days of overlapping records with nearby visibility stations. A total of 1110 sites in the United States are selected for this study.

### 2.2.2 PM$_{2.5}$ Data in Canada

The hourly PM$_{2.5}$ data for Canada from 1995 to 2022 are sourced from the National Air Pollution Surveillance (NAPS) program, which are available at https://www.canada.ca. The NAPS program is a collaborative effort between the Environment and Climate Change Canada and provincial, territorial, and regional governments and is the primary source of environmental air quality data. Since 1984, PM$_{2.5}$ concentrations have been measured in Canada using a dichotomous sampler. Continuous or real-time particle monitoring began in the NAPS network in 1995 using TEOM and beta-attenuation monitoring (Demerjian, 2000). The samples are supplemented by EPA FRM samples obtained after 2009 (Dabek-Zlotorzynska et al., 2011). The number of instruments is growing rapidly, with 410 sites in 2022. A total of 304 PM$_{2.5}$ monitoring sites in Canada are selected for this study.

### 2.2.3 PM$_{2.5}$ Data in Europe

The hourly PM$_{2.5}$ data for Europe from 1998 to 2012 are obtained from the AirBase database, which is available at https://european-union.europa.eu. The hourly PM$_{2.5}$ verified data (E1a) from 2013 to 2022 are obtained from the AirQuality database, which is available at https://www.eea.europa.eu. AirBase is maintained by the European Environment Agency (EEA) through its European Topic Center on Air Pollution and Climate Change Mitigation. Airbase contains air quality monitoring data and information submitted by participating countries throughout Europe. After the Air Quality Directive 2008/50/EC was enforced, the PM$_{2.5}$ data began to be stored in AirQuality database. The main monitoring methods for PM$_{2.5}$ include TEOM and beta attenuation (Green and Fuller, 2006; Chow et al., 2008). The sites are distributed across rural, rural-near city, rural-regional, rural-remote,



suburban, and urban areas. We merge the two datasets with the same site identifiers, and 834 sites
in Europe are selected for this study.

**2.2.4 PM$_{2.5}$ Data in China**

The hourly PM$_{2.5}$ data for China from 2014 to 2022 are obtained from the China National
Environmental Monitoring Center, which are available at https://www.cnemc.cn. China established
air quality monitoring in 1980; 74 cities were the first to publicly release real-time PM$_{2.5}$ in 2013,
and there were more than 1800 air quality observation sites as of 2000 (Su et al., 2022). PM$_{2.5}$
concentrations are measured using the TEOM and beta-attenuation method (Zhao et al., 2016b;
Miao and Liu, 2019). According to the China Environmental Protection Standards, instrument
maintenance, data transmission, data assurance and quality control ensure the reliability of PM$_{2.5}$
concentration measurements. The uncertainty in the PM$_{2.5}$ mass concentration is <5 μg/m$^{-3}$ (Pui et
al., 2014). In this study, a total of 1110 PM$_{2.5}$ monitoring sites are selected.

**2.2.5 PM$_{2.5}$ Data in India**

The hourly PM$_{2.5}$ data for India from 2010 to 2022 are obtained from the Central Pollution Control
Board (CPCB), which are available at https://app.cpcbccr.com. The Air (Prevention and Control of
Pollution) Act of 1981 was enacted by the Central Pollution Control Board (CPCB) of the Ministry
of Environment, Forest and Climate Change (MoEFCC). A standard of 60 μg/m$^3$ PM$_{2.5}$
concentration over 24 hours was added in 2009. The methods used by the Indian National Ambient
Air Quality Standards (NAAQS) for PM$_{2.5}$ and related component measurements include the TEOM,
FRM and FEM (Pant et al., 2019). The measurement precision is ± (1-2) μg/m$^3$ (hour). The National
Air Quality Monitoring Programme (NAMP) is a key air quality monitoring programme employed
by the Government of India, which is managed by the CPCB in coordination with the State Pollution
Control Boards (SPCBs) and UT (union territory) Pollution Control Committees (PCCs). There
were 703 PM$_{2.5}$ monitoring stations as of 2018. Most of these stations (residential and industrial)
are located in urban areas, and others are located sparsely in rural areas. A total of 206 PM$_{2.5}$
monitoring sites are selected for this study.

**2.3 Visibility and Meteorological Data**

The hourly meteorological data from 1959 to 2022 are collected from airport weather observations,
which are available at https://www.weather.gov/asos. Automated observation minimizes the errors
associated with human involvement in data collection, processing, and transmission. The data are
extracted from the Meteorological Terminal Aviation Routine Weather Report (METAR). The World
Meteorological Organization (WMO) sets guidelines for METAR reports, including report format,
encoding, observation instruments and methods, data accuracy, and consistency. These requirements
ensure the consistency and comparability of METAR reports globally. Visibility is a quantity that
describes the atmospheric transparency, usually observed by automated sensors (scattering and
transmission). More than 1000 stations are from the Automated Surface Observing System (ASOS)
in the United States, and other data are sourced from airport reports worldwide. The forward-scatter
visibility sensors at a wavelength of 550 nm for ASOS are consistent with the National Weather
Service of the United States standard transmissometer, with more than 80% of the data within the
limit of ±0.4 km when visibility is less than 2 km (Noaa et al., 1998).
Visibility is an essential variable employed in this study, as research has shown that its reciprocal is



directly proportional to the aerosol extinction coefficient (Wang et al., 2009), which is closely
related to the PM$_{2.5}$ concentration. Considering that temperature, wind speed, wind direction,
humidity, and precipitation are factors that impact particle dispersion, particle growth, and
secondary generation influenced by humidity, as well as the cleansing effect of precipitation (Zhang
et al., 2020), temperature, dew point temperature, temperature-dew point difference, relative
humidity, sea-level pressure, wind speed and direction, precipitation, and sky conditions are also
employed in this study.

### 2.4 Data Preprocessing

The following data preprocessing steps are performed: remove the records with missing visibility,
temperature, dew point temperature, temperature-dew point difference, relative humidity, sea-level
pressure, wind speed, and wind direction data and remove records with hourly precipitation greater
than 0.1 mm, sky conditions marked as 'VV', and relative humidity greater than 90%. Since PM$_{2.5}$
exhibits hygroscopic growth, we calculated the dry visibility for relative humidity values between
30% and 90% (Yang et al., 2021).
$$VISD = VIS/(0.26 + 0.4285 * log(100 - RH))$$

where VIS is the visibility, RH is the relative humidity, and VISD is the dry visibility.
The maximum hourly PM$_{2.5}$ concentration is set to 1000 μg/m$^3$. At least three hourly daily records
are needed. The harmonic mean is used to calculate the daily VIS and daily VISD because it can
better capture rapid weather changes and enhance daily representativeness (Noaa et al., 1998). The
arithmetic average is used for other variables.

### 2.5 Data for Comparison

In this study, our data are compared with other datasets, including two PM$_{2.5}$ datasets based on
satellite AOD data and two reanalysis datasets.

### 2.5.1 ACAG Dataset

The monthly global PM$_{2.5}$ dataset (version V5.GL.04) from 1980 to 2022, with a spatial resolution
of 0.1°, is available from the Atmospheric Composition Analysis Group (ACAG) of Washington
University in St. Louis (https://sites.wustl.edu/acag/datasets/surface-pm2-5/) (Van Donkelaar et al.,
2021). The ACAG PM$_{2.5}$ concentrations are estimated based on satellite (MODIS, VIIRS, MISR
and SeaWiFS) AOD and global vertical aerosol profiles from the Cloud-Aerosol Lidar and Infrared
Pathfinder Satellite Observation (CALIPSO) satellites. The AOD of GEOS-Chem is used to
simulate the spatiotemporally varying geophysical relationship with PM$_{2.5}$. Ground-based PM$_{2.5}$
values are incorporated at a monthly timescale using geographically weighted regression (Van
Donkelaar et al., 2016; Hammer et al., 2020; Van Donkelaar et al., 2021). The coefficients of
determination (R$^2$) for the monthly mean and monitor-based PM$_{2.5}$ concentrations are 0.86 (January),
0.81 (April), 0.72 (July), and 0.78 (October). The R$^2$ with WHO-collocated monitors is between
0.88 and 0.93. The EMSE is between 8 and 13.3 μg/m$^3$.

### 2.5.2 CHAP Dataset

The monthly PM$_{2.5}$ dataset of China High Air Pollutants (CHAP) from 2000 to 2021 is a product
with coverage over China, with a spatial resolution of 1 km, which is available at



https://zenodo.org/records/6398971. The CHAP $PM_{2.5}$ concentration is estimated based on the
MODIS Collection 6 MAIAC AOD product and meteorological variables, surface conditions,
pollutant emissions, and population distributions using a space-time extra-trees model. The $R^2$ and
RMSE of the monthly $PM_{2.5}$ concentration are 0.92-0.94 and ~5.1-10.0 μg/m$^3$, respectively, from
2013 to 2018 (Wei et al., 2020b; Wei et al., 2021).

**2.5.3 MERRA-2 Dataset**

The monthly $PM_{2.5}$ dataset of Modern-Era Retrospective Analysis for Research and Applications
version 2 (MERRA-2) from 1980 to 2022 is a NASA reanalysis dataset with a spatial resolution of
0.5×0.625° and uses the Goddard Earth Observing System version 5 (GEOS-5) coupled to the
Goddard Chemistry Aerosol Radiation and Transport (GOCART) model, which is available at
https://gmao.gsfc.nasa.gov. The aerosol data of GOCART include dust, sea salt, sulfate, black
carbon, and organic carbon, and there are 72 vertical layers from the surface to more than 80 km
altitude. MERRA-2 $PM_{2.5}$ is a dataset produced by the GEOS-5 atmospheric model and data
assimilation system and the three-dimensional variational data analysis (3DVAR) Grid-point
Statistical Interpolation (GSI) meteorological analysis scheme (Randles et al., 2017). In the aerosol
model (GOCART), a $SO_2$ emission database of volcanic material for secondary sources is included.
Aerosol hygroscopic growth depends on the simulated relative humidity. The monthly scale biomass
burning inventory is from RETROv2 from 1980 to 1996; the monthly $SO_2$, $SO_4$, POM, and BC
emissions are from GFEDv3.1 from 1997 to 2009; and the daily scale data are from QFED 2.4-r6
after 2010. The annual anthropogenic $SO_2$ is from EDGARv4.2 between 100 and 500 m above the
surface from 1980 to 2008. The annual Anthropogenic $SO_4$, BC, and POM concentrations are
obtained from AeroCom Phase II from 1980 to 2006. In assimilation systems, satellite AOD
retrievals are used, including AVHRR (over the oceans) from 1998 to 2002, MISR from 2000 to
2014, MODIS Aqua since 2002, and MODIS Terra since 2000 (Buchard et al., 2017; Randles et al.,
2017). The direct observations of the AOD AERONET station from 1999 to 2014 are also
assimilated.
The surface $PM_{2.5}$ concentration in MERRA-2 can be computed using the concentrations of black
carbon [BC], organic carbon [OC], dust [$DUST_{2.5}$], sea salt [$SS_{2.5}$], and sulfate [$SO_4$] (Provençal et
al., 2017) and is expressed as follows (please refer to
https://gmao.gsfc.nasa.gov/reanalysis/MERRA-2/FAQ/#Q4):
$[PM_{2.5}] = [DUST_{2.5}] + [SS_{2.5}] + [BC] + 1.6 \times [OC] + 1.375 \times [SO_4]$.
In this study, we conduct spatiotemporal matching between MERRA-2 $PM_{2.5}$ and the estimated
$PM_{2.5}$.

**2.5.4 CAMS Dataset**

The Copernicus Atmosphere Monitoring Service (CAMS) reanalysis is the latest global reanalysis
dataset of atmospheric composition produced by the European Centre for Medium-Range Weather
Forecasts (ECMWF). We use the single-level monthly $PM_{2.5}$ product from the CAMS reanalysis
from 2003 to 2022, which is available at
https://ads.atmosphere.copernicus.eu/cdsapp#!/dataset/cams-global-reanalysis-eac4. The resolution
is 0.75°. The CAMS reanalysis builds on the experience gained during the earlier Monitoring
Atmospheric Composition and Climate (MACC) reanalysis and CAMS interim reanalysis (Inness




et al., 2019). The ECMWF's Integrated Forecast System (IFS) aerosol and chemistry modules are
applied, and more details on the modules are provided in (2015). The data at 60 model levels are
interpolated to 25 pressure levels. Anthropogenic emissions are from the MACCity inventory from
1960 to 2010 (Granier et al., 2011). The emissions of anthropogenic SOAs are estimated from
MACCity CO emissions. The monthly biogenic emissions of the chemical species are from
MEGAN2.1 (Guenther et al., 2006). The natural $NO_2$ emissions from soils and oceans are obtained
from the Precursors of Ozone and Their Effects in the Troposphere (POET) database for 2000. Daily
biomass burning emissions are from the Global Fire Assimilation System version 1.2 (GFASv1.2)
(Kaiser et al., 2012). More details regarding emissions are provided in Granier (2011). The
incremental 4D-Var data assimilation system is used for the CAMS reanalysis, and the total aerosol
mixing ratio of the single species is derived from the assimilation of satellite retrievals (Benedetti
et al., 2009). The AODs from satellite retrievals are assimilated, including those from AATSR
Envisat from 2002 to 2012 and those from MODIS Terra and Aqua since 2002. For additional
information, please refer to Inness et al. (2019).
The surface $PM_{2.5}$ concentration is estimated by the air density [ρ], sea salt [$SS_{1,2}$], dust [$DD_{1,2,3}$],
nitrate [$NI_{1,2}$], organic matter [OM], black carbon [BC], ammonium [AM], and sulfate [$SO_4$] and is
expressed as follows (Inness et al., 2019):
$[PM_{2.5}] = ρ ×([DD_1] + [DD_2] + [SS_1/4.3] + [0.5 ×SS_2/4.3] + [0.7 × (AM + OM + 0.7NI_1 + SO_4)] +$
$[BC] + 0.25 ×[NI_2])$.
**2.6 Decision Tree Regression**
We employ decision tree regression using the CART algorithm (Teixeira, 2004) to estimate daily
$PM_{2.5}$ concentrations. The key to decision tree regression is to find the optimal split variable and
optimal split point. The optimal split point of the predictor is determined by the minimum mean
squared error, which determines the optimal tree structure. Decision tree regression is a commonly
used nonlinear machine learning method that partitions the feature space based on the mapping
between feature attributes and response values, with each leaf node representing a specific output
for each feature space region. It's ability to handle complex relationships with relatively few model
parameters is advantageous, minimizing the risk of overfitting and enabling the prediction of
continuous and categorical predictive variables.
The predictor includes 11 variables: the reciprocal of dry visibility (Vis_Dry_In), the reciprocal of
visibility (Vis_In), temperature (Temp), dew point temperature (Td), temperature-dew point
difference (Temp-Td), relative humidity (RH), sea-level pressure (SLP), wind speed (WS), wind
direction (WD), numerical time (DateTime) and daily record number (DailyObsNum). The response
variable is the daily observed $PM_{2.5}$ concentration.
We randomly select 80% of the sample data to establish the decision tree regression model, and the
remaining 20% of the sample data are used to test the model's predictive ability. To obtain a stable
model, a 10-fold cross-validation method (Browne, 2000) is used to train the model.
**2.7 Evaluation Metrics**
**2.7.1 Statistical Metrics**
We use the root mean squared error (RMSE), mean absolute error (MAE), and correlation



coefficient (ρ) as evaluation metrics to evaluate the model's performance and predictive ability. The
formulas are given as follows:
$$MSE = \sqrt{\frac{1}{n}\sum_{i=1}^{n}(y_i - \hat{y}_i)^2}$$

$$MAE = \frac{1}{n}\sum_{i=1}^{n}|y_i - \hat{y}_i|$$

$$\rho = \frac{\sum_{i=1}^{n}(y_i - \bar{y})(\hat{y}_i - \bar{\hat{y}})}{sqrt(\sum_{i=1}^{n}(y_i - \bar{y})^2 \sum_{i=1}^{n}(\hat{y}_i - \bar{\hat{y}})^2)}$$

where $y_i$ and $\bar{y}$ are the predicted value and the average of the predicted values. $\hat{y}_i$ and $\bar{\hat{y}}$ are
the target and the average of the target. $i = 1,2,\ldots,n$. $n$ is the length of sample.

### 2.7.2 Partial Dependence

The importance of predictor variables is assessed via partial dependence. Partial dependence
represents the relationship between the individual predictive variable and the predicted response
(Friedman, 2001). By marginalizing the other variables, the expected response of the predicted
variable is calculated. All the partial dependences of the predicted response on the subset of
predicted variables are calculated. The calculation process of the partial dependency method is
described as follows:
The dataset of the predictor is X, $X = [X^1, X^2,\ldots, X^n]$, and n represents the number of predictive
factors. The complement of subset $X^s$ is $X^c$, where $X^s$ is a single variable in X and $X^c$ is all
other variables in X. The predicted response f(x) depends on all variables in X, and it is expressed
as follows:
$$f(x) = f(X^s, X^c)$$

The partial dependence of the predicted response to $X^s$ is expressed as follows:
$$f^s(X^s) = \int f(X^s, X^c)pC(X^c)dX^c$$

where $pC(X^c)$ is the marginal probability of $X^c$, that is, $pC(X^c) \approx \int f(X^s, X^c)dX^s$. Assuming
that the likelihood for each observation is equal, the dependence between $X^s$ and $X^c$ and the
interactions of $X^s$ and $X^c$ in response are not strong. The partial dependence is shown below:
$$f^s(X^s) \approx \frac{1}{N}\sum_{i=1}^{N} f(X^s, X_i^s)$$

where N is the number of observations and $i$ represents the $i$th observation.

### 2.7.3 Mean Center



The mean center is a geostatistical method used to describe the average position of a set of
geographical coordinates. It represents the central tendency of a set of geographical data and aids in
understanding the overall distribution and trends in the dataset. The mean center of the PM$_{2.5}$
concentration shows the overall trend and variability in PM$_{2.5}$. If the mean center is located at the
edge of the dataset, the data distribution is dispersed. Conversely, if the mean center is located at
the center of the dataset, the data distribution is concentrated. This may be relevant for aspects, such
as population distribution, urban development, and economic activities. It is particularly helpful in
understanding the spatial patterns of PM$_{2.5}$. The expression is given as follows:
$$x_{ct} = \sum_{i=1}^{N} c_i * x_i / \sum_{i=1}^{N} c_i$$

$$y_{ct} = \sum_{i=1}^{N} c_i * y_i / \sum_{i=1}^{N} c_i$$

where $x_{ct}$ and $y_{ct}$ represent the longitude and latitude of the mean center, respectively, and $c_i$
represents the PM$_{2.5}$ concentration at the $i$-th site ($x_i$, $y_i$).
**2.7.4 Standard Deviation Ellipse**
The standard deviation ellipse (SDE) is used in statistics and geography to describe the variability
and correlation of multivariate data. The SDE is calculated based on the mean and covariance matrix
of the data (Gong, 2002). This variable shows the dispersion and correlation of the data across
different dimensions. The center of the ellipse corresponds to the mean of the data, while the shape
and size of the ellipse reflect the variability in the data in different directions.
We calculate the SDE using the locations and concentration measurements associated with the PM$_{2.5}$
points. The major axis of the ellipse indicates the primary direction of data variation. The shape and
size of the ellipse reflect the spatial dispersion of the PM$_{2.5}$ concentration. A larger ellipse indicates
greater variability in the PM$_{2.5}$ concentration distribution, while a smaller ellipse denotes a more
concentrated distribution. A circular ellipse indicates little or weak spatial correlation among PM$_{2.5}$
concentrations. A flattened ellipse indicates a spatial correlation between PM$_{2.5}$ concentrations.
**3. Results and Discussion**
**3.1 Evaluation of Variable Importance**
We analyze the influence of predictive variables over the predicted response. The predictive variable
with the highest partial dependence value is the most important predictive variable in the model.
The partial dependence of the predicted response on each predictive variable is calculated for every
model. Figure 2 (a) shows the ranking results of the importance of all the predictive variables. The
variable with the highest dependence on the predicted response is Vis_Dry_In, and the second
highest dependence is Vis_In. The dependence of the predicted response on Temp, Td, Temp-Td,
RH, WS, and wind WD is moderate. The predictive variables with lower dependence include SLP,
DateTime and DailyObsNum.
We count the frequency and proportion of the most important variables in all the models, as shown
in Figure 2 (b). Vis_Dry_In is the most important variable at 2600 sites, contributing 64.8%. Vis_In
was the second most important variable at 575 sites, accounting for 14.3%. This finding indicates
that visibility is the most crucial variable, with a percentage of 79.1%. Temp and Td contribute 6.7%
and 3.5%, respectively. The contribution of other variables combined is 10.7%. The percentages of
the second most important predictive variable are 25.4% for Vis_In, 39.6% for Vis_Dry_In, 14.6%
for Temp, 7.1% for Td and 3.4% for Temp-Td. Among the three most important variables, the
proportions of Temp and Td are 15.7% and 14.3%, respectively.
The results indicate a strong correlation between the PM$_{2.5}$ concentration and visibility, as visibility
can be considered an indicator of air quality without fog or precipitation. Meteorological factors
influence the dispersion and deposition of PM$_{2.5}$ (Gui et al., 2020; Zhong et al., 2022). Temperature
and dew play secondary roles, and other meteorological predictive variables play lesser roles in the
model. Although the number of daily records and time have the most negligible impacts on the PM$_{2.5}$
concentration in the model, they have significant impacts on the cyclical changes and daily
representativeness of PM$_{2.5}$ (Wang et al., 2012; Zhang et al., 2020).

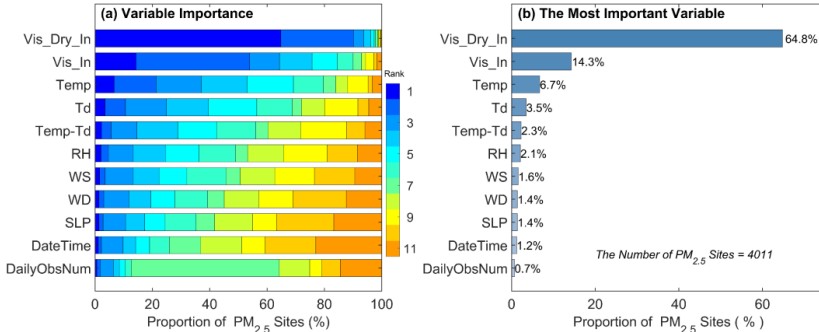


**Figure 2** The importance of predictive variables. The stacked bar (a) shows the importance rankings
of the predictive variables ('rank=1' represents the most important variable). The bar (b) shows the
percentage of the most important predictive variable. The predictive variables are the reciprocal of
dry visibility (Vis_Dry_In), reciprocal of visibility (Vis_In), temperature (Temp), dew point
temperature (Td), temperature-dew point difference (Temp-Td), relative humidity (RH), sea level
pressure (SLP), wind speed (WS), wind direction (WD), numerical time (DateTime) and daily
record number (DailyObsNum). The total number of PM$_{2.5}$ sites is 4011.
**3.2 Evaluation of Model Performance**
**3.2.1 For All Data**
We analyze the linear fitting relationship between all estimated and corresponding response values
to evaluate the model's performance. Figure 3 shows the density scatter plot of the monitored PM$_{2.5}$
concentration (response values) and the estimated PM$_{2.5}$ concentration (estimated values). There is
a total of 8,680,796 data pairs for all the sites. The linear regression coefficient is 0.946 ±0.0002
within the 95% confidence interval, the R$^2$ is 0.95, the RMSE is 7.0 μg/m$^3$, and the MAE is 3.1
μg/m$^3$.

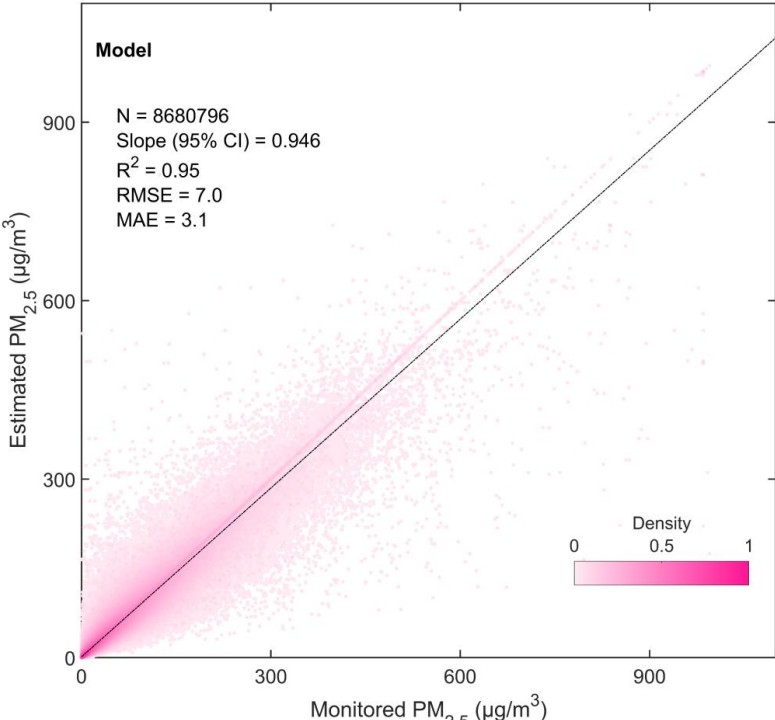


**Figure 3** Density scatter plot (a) between estimated values (estimated PM$_{2.5}$) and the corresponding
response values (monitored PM$_{2.5}$) at the daily scale. The dashed black line is the linear regression
line. N is the length of the data pairs, and Slope is the linear regression coefficient within a 95%
confidence interval (CI). R$^2$ is the coefficient of determination, RMSE is the root mean square error,
and MAE is the mean absolute error.

**3.2.2 For the Site and Region Scales**

We evaluate the model's performance using the RMSE, MAE, and ρ of the estimated and response
values at the site and region scales. Figure 4 shows the spatial distribution (a-c) and frequency
distribution (d-f) of the model's RMSE, MAE, and ρ at all sites. Table 1 lists the model's performance
metrics for all sites and sites in the United States, Canada, Europe, China, and India.

For all sites, the average RMSE is 7.42 μg/m$^3$, with a median of 4.97 μg/m$^3$. The RMSE of 80% of
the sites is less than 11.95 μg/m$^3$. The ratio of the RMSE to the average PM$_{2.5}$ concentration is 29.2%.
The average MAE is 4.01 μg/m$^3$, with a median of 2.66 μg/m$^3$. The MAE is less than 6.62 μg/m$^3$
for 80% of the sites. The MAE-to-mean ratio is 15.8%. The average ρ is 0.90, and the median is
0.91. The ρ of 80% of the sites is greater than 0.87. Previous studies have shown that for PM$_{2.5}$
retrieved from daily visibility or satellite AOD data, the R$^2$ range of the model is from 0.42 to 0.89,
and the RMSE range is from 9.59 μg/m$^3$ to 32.09 μg/m$^3$ (Shen et al., 2016; Liu et al., 2017; Wei et
al., 2019b; Gui et al., 2020; Li et al., 2021; Zhong et al., 2021). This finding indicates that our model





performs well at the daily scale.
At the regional scale, the average RMSE values for the United States, Canada, Europe, China, and
India are 78, 2.86, 4.63, 11.62, and 18.73 μg/m³, respectively, and the mean $PM_{2.5}$ concentrations
are 31.2%, 40.9%, 33.0%, 28.0%, and 27.9%, respectively. The average MAEs for the United States,
Canada, Europe, China, and India are 1.42 μg/m³, 1.36 μg/m³, 2.45 μg/m³, 6.48 μg/m³, and 9.56
μg/m³, respectively; these values correspond to 15.9%, 19.4%, 17.5%, 15.6%, and 14.2%,
respectively, of the mean $PM_{2.5}$ concentration. The average correlation coefficients for the United
States, Canada, Europe, China, and India are 0.88, 0.88, 0.89, 0.92, and 0.92, respectively.
The values of RMSE and MAE are the largest in India. The RMSE is the smallest in the United
States, and the MAE is the smallest in Canada. The ratios of the RMSE and MAE to the mean are
larger in Canada and Europe than in other regions and smaller in China and India than in other
regions. Although the $PM_{2.5}$ concentration varies among regions, the MAE-to-mean concentration
ratio remains at approximately 16%. This finding demonstrates the stability and reliability of the
model.
**Table 1** The results of the model's performance metrics for all sites and sites in the United States
(the US), Canada, Europe, China and India.

| Model | RMSE (μg/m³) | MAE (μg/m³) | ρ (Pearson's correlation) | Mean (μg/m³) | RMSE/Mean (%) | MAE/Mean (%) |
|---|---|---|---|---|---|---|
| All | 7.42 | 4.01 | 0.90 | 25.4 | 29.2 | 15.8 |
| the US | 2.78 | 1.42 | 0.88 | 8.9 | 31.2 | 15.9 |
| Canada | 2.86 | 1.36 | 0.88 | 7.0 | 40.9 | 19.4 |
| Europe | 4.63 | 2.45 | 0.89 | 14.0 | 33.0 | 17.5 |
| China | 11.62 | 6.48 | 0.92 | 41.5 | 28.0 | 15.6 |
| India | 18.73 | 9.56 | 0.92 | 67.0 | 27.9 | 14.2 |

Earth System
Science
Data

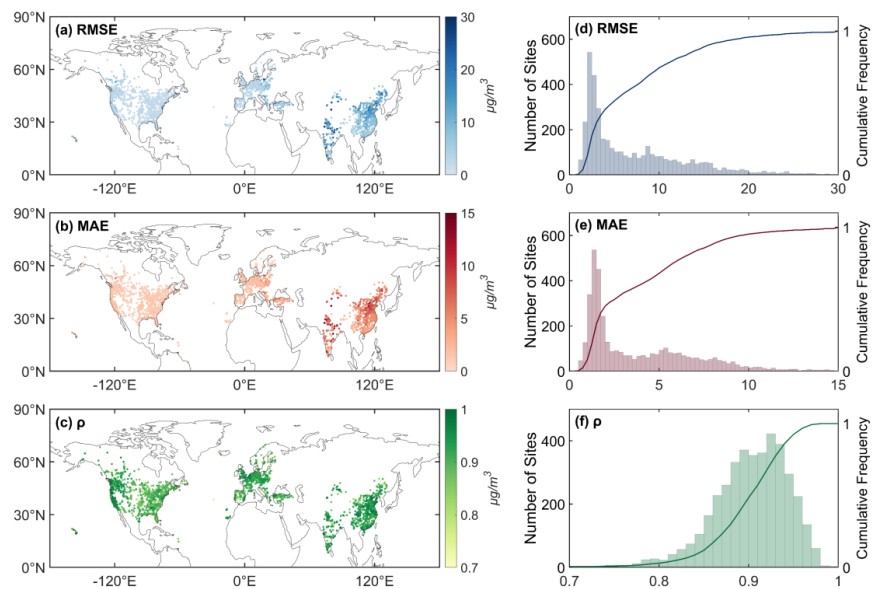

**Figure 4** Spatial distribution (a-c) of the root mean squared error (RMSE), mean absolute error (MAE), and correlation coefficient (ρ) between the model's estimated values and response values. Number of sites (bar) and cumulative frequency (curve) (d-e) of the RMSE, MAE, and ρ.

### 3.2.3 Dependence on the Distance between the $PM_{2.5}$ Site and the Visibility Station

Although the previous analysis elucidates the stability and predictive capability of the model, it is necessary to understand the potential impact of the distance between $PM_{2.5}$ monitoring sites and visibility stations on the model. Most $PM_{2.5}$ monitoring sites are in urban areas, resulting in a relatively concentrated spatial distribution. Visibility stations are strategically placed to capture the characteristics of meteorological factors and have relatively uniform spatial distributions. Consequently, visibility stations and $PM_{2.5}$ monitoring sites are often not collocated, resulting in a certain spatial distance between them. Therefore, we consider the impact of the distance between sites on the model's performance.

Figure 5 shows the relationship between the model performance (ρ and RMSE) and the distance between the visibility stations and the $PM_{2.5}$ monitoring sites. The average distance between all sites is 0.964°, and the correlation coefficient between the model's RMSE and distance is 0.44, which is a moderate correlation. The average ρ of 3786 sites (within a distance of 3°) is 0.90, and the average RMSE is 7.13 μg/m³. The RMSE values of 471 sites are greater than twice the average RMSE of all sites; however, their average ρ (0.91) is greater than the average of all sites. This finding indicates that the model's performance decreases as the distance increases.

For the United States, the average distance is 0.29°. The distance between the 919 (82.8%) sites was less than 0.5°, with ρ and RMSE values of 0.88 and 2.7 μg/m³, respectively. The ρ and RMSE of the 191 sites (more than 0.5°) are 0.88 and 3.1 μg/m³, respectively. The performance of the model is not significantly related to distance.



For Canada, 212 (69.7%) sites have distances of less than 0.5°, with $\rho$ and RMSE values of 0.89
and 2.6 $\mu g/m^3$, respectively. The $\rho$ and RMSE for 92 sites (more than 0.5°) are 0.87 and 3.3 $\mu g/m^3$,
respectively. The correlation coefficient between the RMSE and the distance is 0.33, and the
correlation coefficient between the $\rho$ and the distance is -0.17. The performance of the model
decreases as the distance increases.
For Europe, 541 (64.8%) sites have distances of less than 0.5°, with $\rho$ and RMSE values of 0.90 and
4.0 $\mu g/m^3$, respectively. The $\rho$ and RMSE of the 293 sites (more than 0.5°) are 0.88 and 5.7 $\mu g/m^3$,
respectively. The correlation coefficient between the RMSE and the distance is 0.19.
For China, 303 (19.5%) sites have a distance of less than 0.5°, with $\rho$ and RMSE values of 0.95 and
9.5 $\mu g/m^3$, respectively. The $\rho$ and RMSE for 1254 sites (more than 0.5°) are 0.91 and 12.1 $\mu g/m^3$,
respectively. The correlation coefficient between the RMSE and the distance is 0.23. The correlation
coefficient between $\rho$ and distance is -0.71. As the distance increases, the correlation coefficient
significantly decreases.
For India, the $\rho$ and RMSE of 117 (56.8%) sites with a distance of less than 0.5° are 0.94 and 18.7
$\mu g/m^3$, respectively. The $\rho$ and RMSE of 89 sites (more than 0.5°) are 0.89 and 18.8 $\mu g/m^3$,
respectively. The correlation coefficient between $\rho$ and distance is -0.36.
The above results indicate no significant correlation between model performance and distance in
the United States and Europe, as these regions have adequate visibility stations. However, in China,
India, and Canada, the performance of models is influenced by distance. Particularly in China, due
to the limited number of visibility stations, although the correlation coefficient decreases with
distance, there is no significant change in the RMSE. The correlation coefficient for visibility
remains near 0.4. Even when the distance between two visibility stations reaches 1000 km, the
maximum correlation coefficient for visibility remains near 0.4 (Fei et al., 2023). To acquire more
$PM_{2.5}$ sample data, we do not disregard these distant sites since the models still shows a good
performance for these sites. Nevertheless, more sufficient visibility stations in the same locations
can enhance the model's performance.

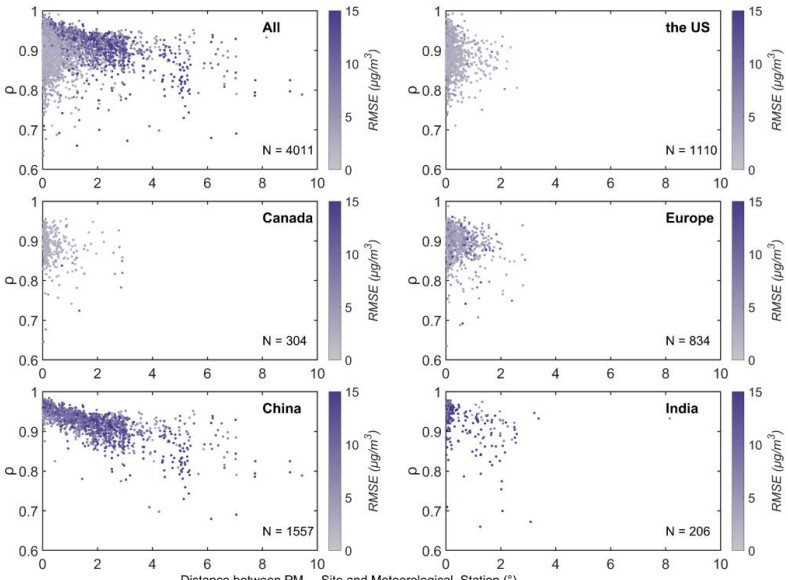


**Figure 5** Scatter plots of the distance between the PM$_{2.5}$ site and visibility station and the model's
correlation coefficient (ρ) for all sites and sites in the United States, Canada, Europe, China, and
India. The color bar represents the root mean square error (RMSE) of the model. N is the number
of sites.
**3.3 Evaluation of Model's Predictive Ability**
**3.3.1 For All Data**
A total of 1,149,152 pairs of test data is employed to evaluate the model's predictive ability. Figure
6 shows the density scatter plot between the predicted PM$_{2.5}$ concentration and the test PM$_{2.5}$
concentration. The results indicate that the linear regression coefficient is 0.862 ± 0.001 within a
95% confidence interval, R$^2$ is 0.80, RMSE is 13.5 μg/m$^3$, and MAE is 6.9 μg/m$^3$. Previous studies
have shown that the R$^2$ range of the model's predictive results at the daily scale is 0.42-0.89, and the
RMSE range is 9.59-32.09 μg/m$^3$ (Gui et al., 2020; Zhong et al., 2021). The test results exhibit
excellent predictive capability.

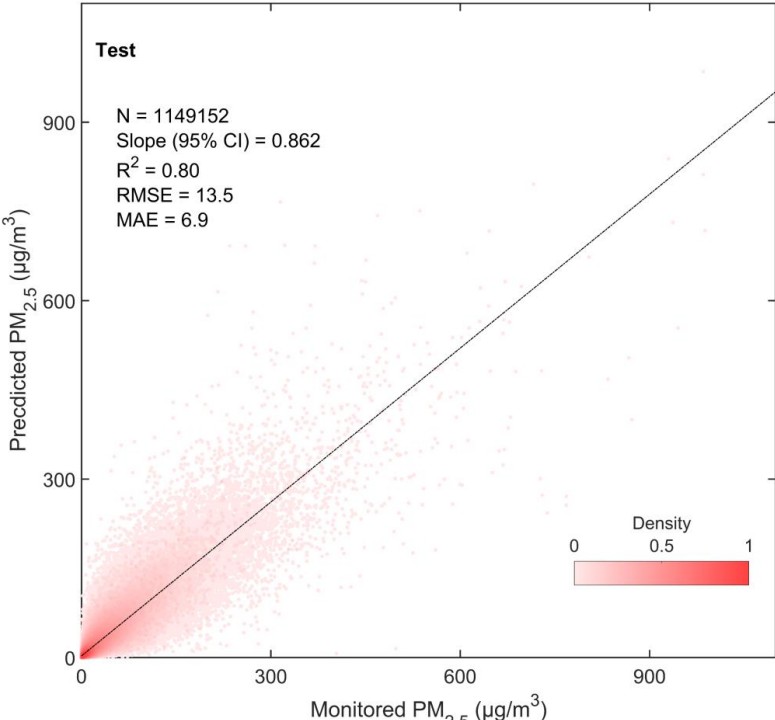


**Figure 6** Density scatter plot (a) between the predicted PM$_{2.5}$ concentration and monitored PM$_{2.5}$
concentration of the test results at the daily scale. The dashed black line is the linear regression line.
N is the length of the data pairs, and Slope is the linear regression coefficient within a 95%
confidence interval (CI). $R^2$ is the coefficient of determination, RMSE is the root mean square error,
and MAE is the mean absolute error.

### 3.3.2 For the Site and Region Scales

We analyze the test results for Canada, the United States, Europe, China, and India to assess the
predictive ability of the model in different regions. Figure 7 shows the spatial distributions of the
test RMSE, MAE, and ρ and their frequency and cumulative frequency distributions. Table 2 lists
the test results of the metrics.

For all sites, the average RMSE is 12.60 μg/m$^3$. The RMSE-to-mean ratio is 48.6%. The average
MAE is 8.52 μg/m$^3$. The MAE-to-mean ratio is 32.9%. The average ρ is 0.77.

For the United States, the RMSE, MAE, and ρ are 4.90 μg/m$^3$, 3.15 μg/m$^3$, and 0.71, respectively.
For Canada, the RMSE, MAE, and ρ are 4.89 μg/m$^3$, 3.01 μg/m$^3$, and 0.74, respectively. The results
in the United States and Canada are better in the west than in the east. The RMSE, MAE, and ρ for
Europe are 7.54 μg/m$^3$, 4.91 μg/m$^3$, and 0.77, respectively. For China, the RMSE, MAE, and ρ are
20.16 μg/m$^3$, 13.81 μg/m$^3$, and 0.81, respectively. For India, the RMSE, MAE, and ρ are 28.84
μg/m$^3$, 19.57 μg/m$^3$, and 0.83, respectively. The results show that in developing regions (China and

Earth System
Science
Data

562 India), ρ is better than that in developed regions (the United States, Canada, and Europe), which
563 means that the predictive ability of the model is better for severely polluted regions.

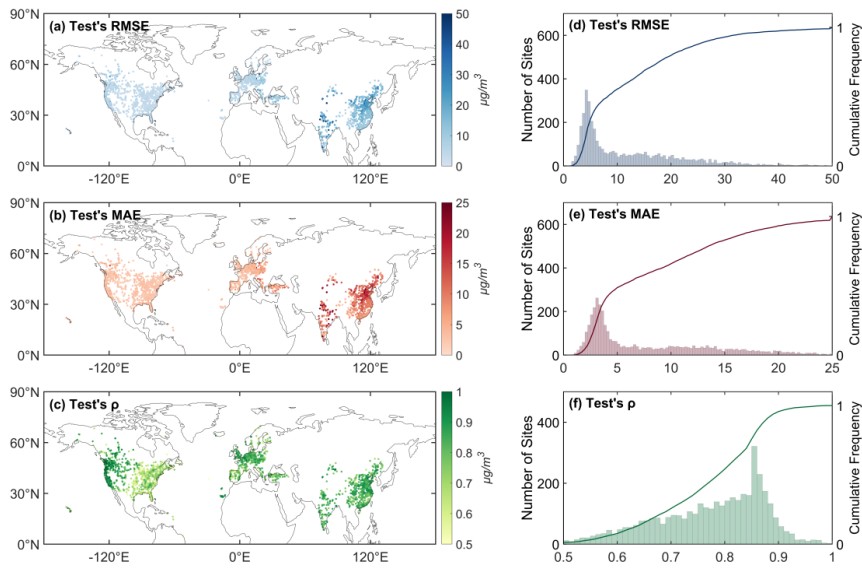

564

**Figure 7** Spatial distribution (a-c) of the root mean squared error (RMSE), mean absolute error
(MAE), and correlation coefficient (ρ) between the model's predicted values and test values.
Number of sites (bar) and cumulative frequency (curve) (d-e) of the RMSE, MAE, and ρ.

**Table 2** The test results of the model's performance metrics for all sites and sites in the United States,
Canada, Europe, China and India.

| Test | RMSE (μg/m³) | MAE (μg/m³) | ρ (Pearson's correlation) | Mean (μg/m³) | RMSE/Mean (%) | MAE/Mean (%) |
|---|---|---|---|---|---|---|
| All | 12.60 | 8.52 | 0.77 | 25.9 | 48.6 | 32.9 |
| America | 4.90 | 3.15 | 0.71 | 9.1 | 53.8 | 34.6 |
| Canada | 4.89 | 3.01 | 0.74 | 7.2 | 67.9 | 41.1 |
| Europe | 7.54 | 4.91 | 0.77 | 14.4 | 52.3 | 34.1 |
| China | 20.16 | 13.81 | 0.81 | 42.2 | 47.7 | 32.7 |
| India | 28.94 | 19.62 | 0.83 | 67.6 | 42.8 | 29.0 |

**3.4 Uncertainties and Limitations**

**3.4.1 Uncertainty in the Pollution Level**

Figure 8 shows the uncertainty in the predicted $PM_{2.5}$ concentration with respect to the pollution
level of the monitored $PM_{2.5}$. For all sites, the uncertainty in the bias increases as the pollution level
increases. The mean bias and the median bias shift from positive to negative with increasing
pollution levels. The mean bias of 88.4% of the data is less than 2 μg/m³. A mean bias of 86.9%
(<40 μg/m³) is positive, and a median bias of 38.9% (<8 μg/m³) is positive. This result indicates that
the model overestimates at low concentrations.
The bias for each region also increases with pollution level. For sites in the United States, the mean
bias of 92.1% is less than 2 μg/m³. A total of 69.1% (<10 μg/m³) are positive. For sites in Canada,
the mean bias of 82.5% is less than 2 μg/m³. A total of 73.3% are positive (<8 μg/m³). Among the
data (<8 μg/m³), 57.9% of the median is positive. For sites in Europe, the mean bias of 64.8% is less
than 2 μg/m³, and 69.8% is positive. A total of 49.0% of the median is positive. For sites in China,
81.8% of the bias is less than 5 μg/m³, and 68.9% (<45 μg/m³) is positive. A total of 48.0% (<30
μg/m³) of the median is positive. For sites in India, 80.5% of the bias is less than 8 μg/m³, and 73.5%
(<80 μg/m³) is positive. A total of 52.6% (<60 μg/m³) of the median values are positive.

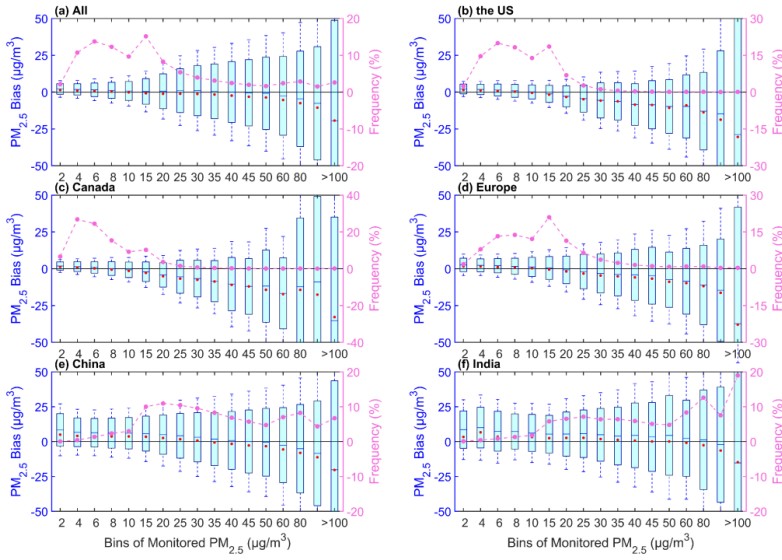


**Figure 8** Boxplots of the pollution level and bias (predicted PM₂.₅ - monitored PM₂.₅) for all sites
(a), sites in the United States (b), Canada (c), Europe (d), China (e), and India (f). The box's upper
and lower limits represent ±1 standard deviation, the whiskers represent 2 times the standard
deviation, the red circle represents the median, and the short line represents the mean bias. The
frequency (%) on the right y-axis represents the percentage of data with different pollution levels
(dashed line).
**3.4.2 Uncertainty in the Station Elevation**
With the spatial variability in PM₂.₅, we analyze the mean bias at different station elevations. Figure
9 shows the relationships between the elevations of the visibility stations and the bias. The bias
exhibits variations across different elevations for all sites. A total of 89.5% of the data are at an
elevation of 1 km. The mean bias ranges from -0.1 to 0.5 μg/m³. High uncertainties in bias occur at
elevations below 0.2 km, 0.4-0.5 km, and 1-3 km. A total of 88.5% of the data have positive mean
biases. Negative biases are observed at elevations of 0.6-0.8 km, 3 km, and 5 km. A total of 57.7%
of the data have a positive median. This finding indicates a nonsignificant overestimation of the
predicted PM$_{2.5}$ concentration due to the various elevations.
The bias patterns vary across regions. For the United States, 92.8% of the data correspond to
elevations below 1 km. The mean bias ranges from -0.1 to 0.5 μg/m$^3$. A total of 88.8% of the mean
biases are positive, and the median of 99% is positive. For Canada, 90.1% of the data correspond to
elevations below 1 km. The mean bias ranges from -0.1 to 0.2. A total of 46.5% of the mean bias is
positive, and the median is positive except at elevations of 0.7 km and 4 km. A higher uncertainty
in the bias occurs at elevations ranging from 0.5-0.8 km. For Europe, 92.9% of the data correspond
to elevations below 1 km. The bias ranges from -0.2 to 0.2 μg/m$^3$. A total of 62.7% of the mean bias
is negative, and the median is positive. High standard deviations are observed at elevations of 0.2
km, 0.05 km, and 0.5-0.6 km. A significant bias occurs at 0.6 km. For China, 81.9% of the data
correspond to elevations below 0.5 km. The median is positive, and the mean bias is positive except
at 0.1 km. The lowest standard deviation occurs at an elevation of 0.3 km. For India, the mean bias
ranges from -0.3 to 0.9 μg/m$^3$. The highest bias occurs at an elevation of 0.3 km. There is a negative
mean bias in the range of 0.1-0.4 km. The medians are positive except at an elevation of 0.4 km.

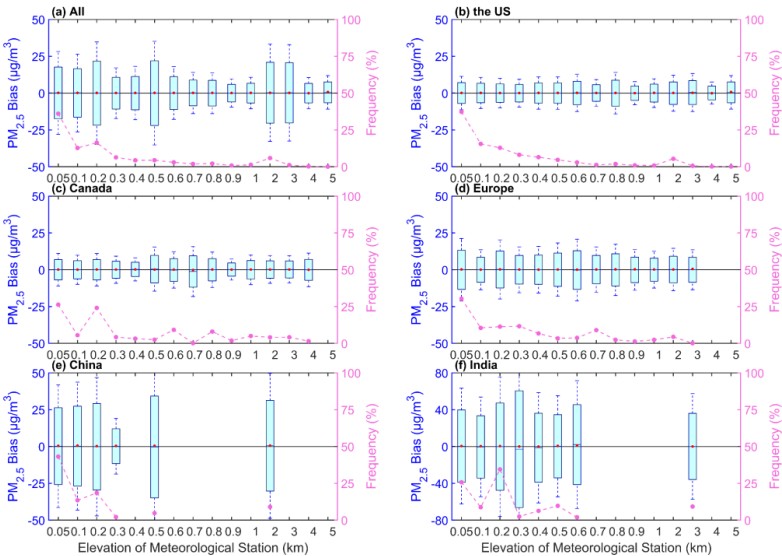


**Figure 9** Boxplots of the elevation of the visibility station and bias (predicted PM$_{2.5}$ - monitored
PM$_{2.5}$) for all sites (a), sites in the United States (b), Canada (c), Europe (d), China (e), and India (f).
The box's upper and lower limits represent ±1 standard deviation, the whiskers represent 2 times the
standard deviation, the red circle represents the median, and the short line represents the mean bias.
The frequency (%) on the right y-axis represents the percentage of data at different pollution levels
(dashed line).
**3.4.3 Uncertainty in the Station Distance**
As the visibility stations and PM$_{2.5}$ sites are not collocated, we analyze the PM$_{2.5}$ mean bias at
different distances. Figure 10 shows the distance between the visibility of the station and the PM$_{2.5}$
site and bias. For all sites, the standard deviation gradually increases with distance, indicating an
increase in uncertainty with increasing distance. Except at distances of 0.05° and 1°, the mean bias
is positive. The median is positive. For each region, the distance of the largest average bias is 3° in
the United States, 3° in Canada, 0.8° in Europe, 10° in China, and 0.4° in India. The distances are
below 1° in the United States, Canada, Europe, and India, while they are 1-3° in China. This finding
is due to the limited number of visibility sites in China. The mean bias exhibits greater uncertainties
in China and India.

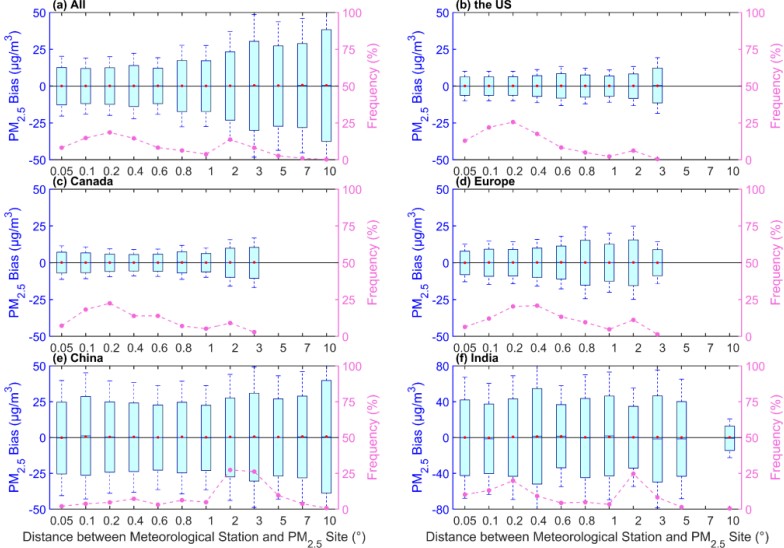


**Figure 10** Boxplots of the distance between the visibility station and the $PM_{2.5}$ site and bias
(predicted $PM_{2.5}$ - monitored $PM_{2.5}$) for all sites (a), sites in the United States (b), Canada (c), Europe
(d), China (e), and India (f). The box's upper and lower limits represent ±1 standard deviation, the
whiskers represent 2 times the standard deviation, the red circle represents the median, and the short
line represents the mean bias. The frequency (%) on the right y-axis represents the percentage of
data under different pollution levels (dashed line).

### 3.4.4 Discussion on the Uncertainties and Limitations

There are some uncertainties and limitations in this study. The upper limit of visibility ($PM_{2.5}$) is 10
km (1000 $\mu g/m^3$), which can cause some uncertainties in modeling. The maximum distance for
spatial matching between the visibility stations and $PM_{2.5}$ monitoring sites is 10° due to the spatial
variability in aerosols, which may increase the uncertainty in the estimated $PM_{2.5}$ concentration. The
boundary layer height is closely related to the vertical structure of $PM_{2.5}$, and reanalysis data may
introduce uncertainty to the model. Because of the nonuniform vertical distribution of aerosols, the
different elevations of the visibility stations and the $PM_{2.5}$ monitoring sites further increase the
uncertainty in estimating $PM_{2.5}$. In addition, the spatial coverage of visibility stations, especially in
China, is limited, which may increase the uncertainty in the representativeness of regional $PM_{2.5}$
trends and pollution levels. With the increasing human concern about air pollution and the
implementation of pollution control measures, the types of major atmospheric pollutants have



changed, the composition of particulate matter has also evolved, the scattering and absorption
characteristics may have changed, and the relationship between visibility and $PM_{2.5}$ may change.
These changes may lead to uncertainty in estimating historical $PM_{2.5}$, especially before 2000
(ground and satellite observations are limited). Despite these limitations, we establish a long-term
$PM_{2.5}$ dataset based on visibility from 1959 to 2022, providing insights into the long-term
spatiotemporal characteristics of $PM_{2.5}$ in the Northern Hemisphere.

**4 Comparisons with Other $PM_{2.5}$ Datasets**

We compare the monthly estimated $PM_{2.5}$ with the $PM_{2.5}$ of those derived from a satellite AOD and
two reanalysis datasets, including (1) ACAG, the monthly satellite-derived $PM_{2.5}$ from 1998 to 2022
(Van Donkelaar et al., 2019; Hammer et al., 2020); (2) MERRA-2, the monthly $PM_{2.5}$ from 1980 to
2022 (Buchard et al., 2016; Buchard et al., 2017; Gelaro et al., 2017); and (3) CAMS, the monthly
$PM_{2.5}$ from 2003 to 2022 (Inness et al., 2019). The time ranges for comparing the estimated $PM_{2.5}$
with the ACAG, MERRA-2, and CAMS data are 1998-2022, 1980-2022, and 2003-2022,
respectively. The monthly average should meet a minimum requirement of at least ten days per
month.

**4.1 Monthly Frequency and Annual Cycle of $PM_{2.5}$**

We compare the frequency of the estimated $PM_{2.5}$ concentration at different pollution levels, with
an interval of 1 μg/m$^3$, with three other datasets. Figure 11 shows the monthly $PM_{2.5}$ frequencies of
the estimated, ACAG, MERRA-2, and CAMS datasets for all sites and regional sites.
Compared with the ACAG data, they exhibit similar frequency distributions. However, the
frequency of estimated $PM_{2.5}$ concentrations is greater at high pollution levels at all sites. Regionally,
the frequency distributions are similar at different pollution levels in the United States and Canada.
In Europe, China, and India, the frequency of high concentrations is greater than that of the ACAG.
Compared with the MERRA-2 data, the frequency distribution of the estimated data is similar to
that of the ACAG for all the sites. Regionally, the frequency distributions of the estimates are
comparable in the United States and Canada. However, in Europe, China, and India, the differences
in the frequency of high pollution levels are greater than those in the ACAG.
Compared with the CAMS data, the frequency distributions at high pollution levels are similar, but
the frequency at high pollution levels is lower. Regionally, Europe differs from other regions, as the
frequency of high pollution levels is higher.

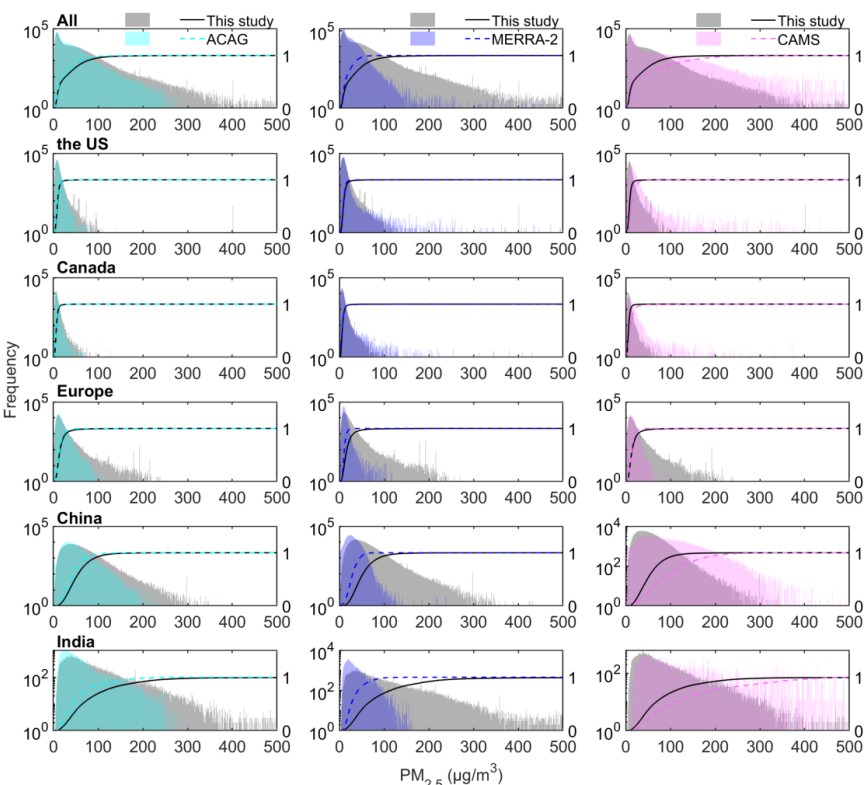

**Figure 11** Frequency (left axis) and cumulative frequency (right axis) of monthly PM$_{2.5}$. The time range of the estimated PM$_{2.5}$ corresponds to the time range of the three datasets (ACAG from 1998 to 2022, MERRA-2 from 1980 to 2022, and CAMS from 2003 to 2022). The bins range from 0 to 500 μg/m$^3$ with an interval of 1 μg/m$^3$.

In Figure 12, we compare the multiyear monthly average PM$_{2.5}$ concentration with that of the three datasets. For all sites, the correlation coefficients between the estimated and ACAG, MERRA-2, and CAMS data are 0.99, 0.42, and 0.93, respectively, and the average biases (average relative biases) are 6.6 μg/m$^3$ (26%), 14.1 μg/m$^3$ (76%), and -19.1 μg/m$^3$ (-37%), respectively. The estimated multiyear average monthly PM$_{2.5}$ concentrations are higher for ACAG and MERRA-2 and lower for CAMS. The correlation coefficient is highest for ACAG and lowest for MERRA-2.

Compared with the ACAG data, the correlation coefficients are 0.97, 0.96, 0.98, 0.99, and 0.99, with average biases (average relative biases) of 0.8 μg/m$^3$ (9%), 0.5 μg/m$^3$ (7%), 2.2 μg/m$^3$ (16%), 10.8 μg/m$^3$ (26%), and 31.4 μg/m$^3$ (62%) in the United States, Canada, Europe, China, and India, respectively. The annual variations in the two datasets are nearly consistent across all regions. The bias is less than 10% for the United States and Canada, while India exhibits the largest bias.

Compared with the MERRA-2 data, the correlation coefficients are 0.30, 0.61, -0.25, 0.80, and 0.45, with average biases (average relative biases) of 1.1 μg/m$^3$ (16%), 0.2 μg/m$^3$ (5%), 7.5 μg/m$^3$ (67%), 24.1 μg/m$^3$ (83%), and 56.1 μg/m$^3$ (169%) in the United States, Canada, Europe, China, and India,

respectively. There are differences in the annual variations between the two datasets, particularly during winter (November to January) and spring (February to March), in all regions. The largest difference occurs in March and September to December in Europe, showing the opposite trend. The highest correlation coefficient is observed in China, which has the second largest bias. The largest bias is in India.

Compared with the CAMS data, the correlation coefficients are 0.29, 0.22, 0.02, 0.91, and 0.98, with average biases (average relative biases) of -5.4 μg/m$^3$ (-34%), -5.0 μg/m$^3$ (-38%), 2.7 μg/m$^3$ (21%), -38.7 μg/m$^3$ (-42%), and -52.7 μg/m$^3$ (-36%) in the United States, Canada, Europe, China, and India, respectively. The annual variations between the CAMS and ACAG data are similar in China and India but have more significant biases. The smallest differences in the United States and Canada occur in January and December. In Europe, the months with more significant biases are January to March and September to December, while biases are smaller in other months.

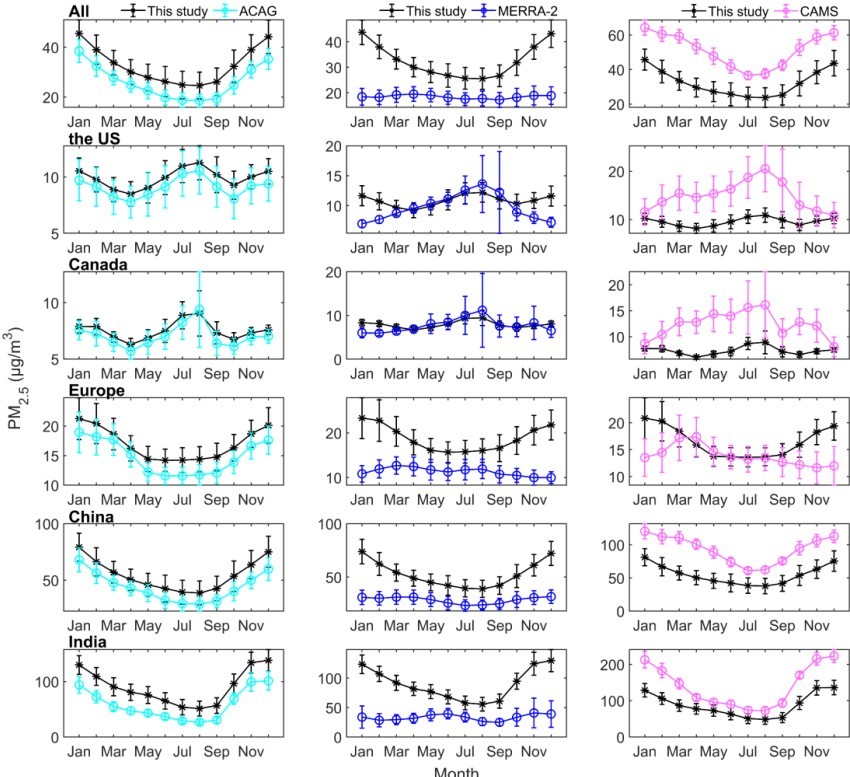

**Figure 12** Multiyear monthly average PM$_{2.5}$ of our data and the three datasets. The time range of the estimated PM$_{2.5}$ corresponds to the time range of the three datasets (ACAG data from 1998 to 2022, MERRA-2 data from 1980 to 2022, and CAMS data from 2003 to 2022).

**4.2 Time Series at the Annual Scale**

Figure 13 shows the annual average PM$_{2.5}$ concentration from 1959 to 2022 in different regions, along with a comparison to the PM$_{2.5}$ concentrations derived from other datasets. Another dataset is



used for comparison in China: the monthly $PM_{2.5}$ of the CHAP from 2000 to 2021 (Wei et al., 2020b;
Wei et al., 2021). We use correlation coefficients, mean bias and mean relative bias to compare the
relationships and differences among the $PM_{2.5}$ datasets.
In the United States, the estimated $PM_{2.5}$ concentrations exhibit correlation coefficients of 0.96, 0.88,
and -0.38 with the ACAG, CAMS, and MERRA-2 data, respectively; the mean bias (mean relative
bias) is 0.8 (10%), -5.4 (-35%), and 1.1 (13%) for each dataset, respectively.
In Canada, the estimated $PM_{2.5}$ concentrations exhibit correlation coefficients of 0.84, 0.62, and -
0.46 with the ACAG, CAMS, and MERRA-2 data, respectively; the mean bias (mean relative bias)
is 0.5 $\mu g/m^3$ (7%), -5.1 $\mu g/m^3$ (-40%), and 0.2 $\mu g/m^3$ (6%) for each dataset, respectively.
In Europe, the estimated $PM_{2.5}$ concentrations exhibit correlation coefficients of 0.96, 0.96, and 0.76
with the ACAG, CAMS, and MERRA-2 data, respectively; the mean bias (mean relative bias) is 2.3
$\mu g/m^3$ (15%), 2.6 $\mu g/m^3$ (20%), and 7.5 $\mu g/m^3$ (66%) for each dataset, respectively.
In China, the estimated $PM_{2.5}$ concentrations exhibit correlation coefficients of 0.78, 0.98, 0.81, and
0.51 with the ACAG, CHAP, CAMS, and MERRA-2 data, respectively; the mean bias (mean
relative bias) is 10.7 $\mu g/m^3$ (24%), 2.5 $\mu g/m^3$ (4%), -39.1 $\mu g/m^3$ (-42%), and 24 $\mu g/m^3$ (90%) for
each dataset, respectively.
In India, the estimated $PM_{2.5}$ concentrations exhibit correlation coefficients of -0.3, -0.02, and -0.09
with the ACAG, CAMS, and MERRA-2 data, respectively; the mean bias (mean relative bias) is
29.9 $\mu g/m^3$ (53%), -58.9 $\mu g/m^3$ (-40%), and 56.1 $\mu g/m^3$ (203%) for each dataset, respectively. From
2013 to 2022, the correlation coefficients with the ACAG and CAMS data are 0.71 and 0.70,
respectively. The trend of visibility declines from 1961 to 2008. The frequency of visibility
(exceeding 10 km) in the afternoon decreases by 46%, and the frequency of visibility (below 4 km)
in the morning increases by 21% (Jaswal et al., 2013), particularly in the central and northern regions.
The low cloud cover significantly increases from 1960 to 2010 in the Indo-Gangetic Plain and the
northwestern and eastern coasts of India (Jaswal et al., 2017). The average total cloud cover is 3.4
okta from 1960 to 2007, with a decrease of 0.07 okta/decade (Jaswal, 2010). However, the indirect
impact of aerosols on cloud formation do not influence cloud cover (Ramanathan et al., 2005). The
prevalence of clouds poses challenges for satellite retrievals in these areas, potentially contributing
to substantial disparities between $PM_{2.5}$ concentrations estimated based on visibility and satellite
retrievals. The CAMS reanalysis data are calibrated using satellite data    and thus show consistency
with the trend in AOD retrievals from satellites; the anthropogenic emission data are from the
MACCity inventory (Inness et al., 2019), and there are significant variations among different
anthropogenic emission inventories, particularly before 2010, which leads to substantial
uncertainties in India (Granier et al., 2011; Liu et al., 2022). These issues exist to a greater or lesser
extent in other regions, which may contribute to the increased disparities between estimated $PM_{2.5}$
and reanalysis data before 2012.

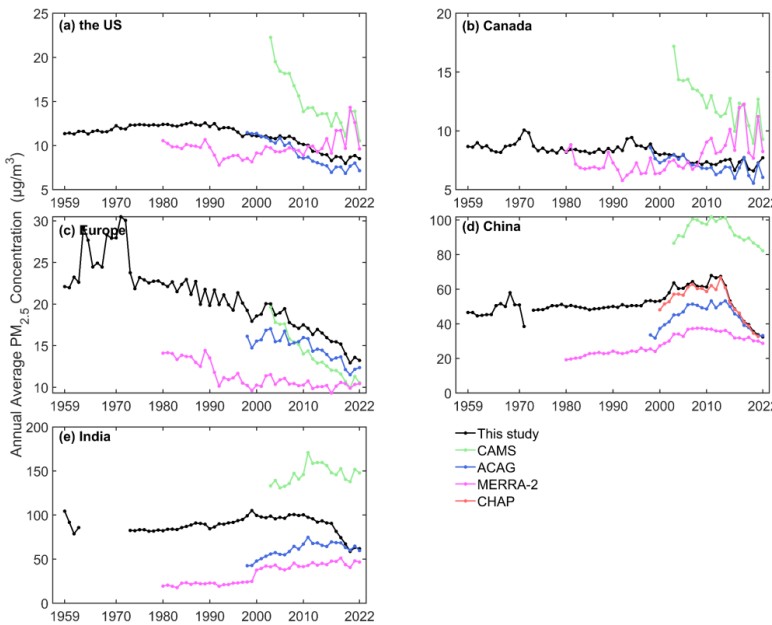

**Figure 13** Annual mean PM$_{2.5}$ concentration from 1959 to 2022 in the United States (US) (a), Canada (b), Europe (c), China (d), and India (e). The other four datasets are ACAG from 1998 to 2022, CHAP from 2000 to 2021, MERRA-2 from 1980 to 2022, and CAMS from 2003 to 2022.

**4.3 Discussion on the Differences among the PM$_{2.5}$ Datasets**

PM$_{2.5}$ is considered a pollutant that decreases visibility. There is a negative correlation between visibility and PM$_{2.5}$ concentration, and the reciprocal of visibility is proportional to the extinction coefficient, which is closely related to the concentration of particulate matter (Wang et al., 2012; Zhang et al., 2017; Zhang et al., 2020). Prior to the widespread implementation of PM$_{2.5}$ measurements or lack of measurement of particulate matter, visibility is often used as a proxy for particulate matter pollution (Huang et al., 2009; Singh et al., 2020). It is the basis for using visibility to estimate PM$_{2.5}$ concentration. Studies have shown that meteorological observations such as temperature and humidity also play an important role in estimating PM$_{2.5}$ concentration using visibility (Shen et al., 2016; Xue et al., 2019; Zhong et al., 2021). The advantages of ground-based visibility and other meteorological variables observations include long-term records, high temporal resolution, and good data completeness, and the visibility observations from airports can be traced back to 1959 in this study. Therefore, we employ a machine learning approach to establish the relationship between PM$_{2.5}$ and visibility and other meteorological variables, and estimate the long-term historical PM$_{2.5}$ concentration from 1959 to 2022, and discuss the limitations and uncertainties. It should be noted that not all sites of PM$_{2.5}$ have the time range from 1959 to 2022, which depends on the record length of matched visibility station.

There are differences between PM$_{2.5}$ based on visibility, PM$_{2.5}$ based on satellite retrievals, and



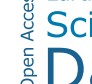

$PM_{2.5}$ of reanalysis. $PM_{2.5}$ based on satellite retrievals typically requires consideration of aerosol
vertical profiles usually provided by observations, assumptions, or chemical transport models to
obtain the aerosol properties near the surface (Van Donkelaar et al., 2010; Wei et al., 2019b; Van
Donkelaar et al., 2021). $PM_{2.5}$ from reanalysis usually requires accurate meteorological fields and
emission inventories. Although ERA5 has provided meteorological reanalysis since 1940, the
historical emission inventories and physical-chemical mechanisms in the chemical transport model
still have significant uncertainties, which increase the uncertainty in particulate matter concentration.
Additionally, the assimilated data in reanalysis mainly consist of satellite AOD and ground-based
AOD, aiming to improve column aerosol properties, without considering near-surface $PM_{2.5}$
(Buchard et al., 2017; Gelaro et al., 2017; Provençal et al., 2017; Huijnen et al., 2019; Inness et al.,
2019; Ali et al., 2022). These factors contribute to the differences in estimating $PM_{2.5}$ concentration
among the three methods.
In this study, the upper limit of the estimated daily $PM_{2.5}$ concentration is set to 1000 μg/m³ because
the $PM_{2.5}$ concentration is greater than 500 during heavy pollution weather, which may contribute
to the higher frequency at high pollution levels than in the other datasets. We do not delete visibility
records during sand and dust weather when preprocessing the data, which may lead to an
overestimation of $PM_{2.5}$ in dusty areas, such as northern China and northwestern India.
The frequency and monthly/annual variations in our data are consistent with those of $PM_{2.5}$ based
on satellite retrievals (ACAG and CHAP). The concentration level is higher than in those datasets
because their upper limits are lower. The AOD is a physical quantity that describes the properties of
aerosol columns. It is important to consider the vertical structure of aerosols when establishing a
connection between AOD and near-ground $PM_{2.5}$. Van Donkelaar et al. (2006; 2010) demonstrated
that aerosol vertical profile errors in chemical transport models and AOD retrieval and sampling
result in an approximately 25% uncertainty of one standard deviation. Sensitivity testing shows that
a 1% estimation error in the AOD can lead to a 0.27% estimation error in the $PM_{2.5}$ concentration
(Wei et al., 2021). Visibility is a near-surface observation that is not affected by clouds or surface
types and has high temporal resolution (Liu et al., 2017; Singh et al., 2020; Zhong et al., 2021). In
section 3.4, the uncertainty analysis provides an explanation for the overestimation.
In section 2.6.3, we introduce the chemical model, emission, and assimilation of MERRA-2. The
$PM_{2.5}$ concentration from MERRA-2 does not include nitrates, and the assimilation of AOD mainly
provides constraints on aerosols after 2000 (Buchard et al., 2016; Randles et al., 2017; Ali et al.,
2022). The lack of nitrate is a limitation in areas with high nitrate concentrations. For example, an
extreme pollution event over China in January 2013 is not captured well (Buchard et al., 2017). Ali
et al. (2022) used $1.4 \times [SO_4^{2-}]$ to represent nitrate concentration, and the results showed a
correlation coefficient of 0.55 with the observed $PM_{2.5}$. Compared to the ACAG over the United
States, which has a low nitrate concentration, the MERRA-2 surface $PM_{2.5}$ concentration is greater
in rural areas than in urban and suburban areas, with high and localized emissions reducing the
representation of the grid mean $PM_{2.5}$ (Buchard et al., 2017). Therefore, the lack of nitrate and
insufficient assimilation data are the key factors leading to the significant differences between the
two datasets.
In section 2.6.4, we introduce the CAMS $PM_{2.5}$. The $PM_{2.5}$ concentration from CAMS is
significantly greater than the estimated $PM_{2.5}$ concentration and follows a similar annual cycle,



except in Europe. In Europe, the CO and $NO_2$ concentrations in CAMS are lower than those in
winter (Flemming et al., 2015), which may lead to the underestimation of nitrate emissions and its
precursors, resulting in the underestimation of $PM_{2.5}$ concentrations. Some studies have reported
similar results (Kong et al., 2021; Ryu and Min, 2021; Ali et al., 2022; Jin et al., 2022). This finding
may be related to the vertical section structure, composition, and microphysical properties of
aerosols (Ali et al., 2022). Because $NO_2$ emissions are obtained by multiplying CO emissions by a
factor of 0.2, the uncertainty in nitrate increases. Studies have shown that the uncertainties in
MACCity (Huijnen et al., 2019) and dust (Ukhov et al., 2020) also cause overestimation in CAMS
$PM_{2.5}$.
Overall, our $PM_{2.5}$ dataset has good consistency with $PM_{2.5}$ based on satellite AOD data. There are
some differences in the reanalysis $PM_{2.5}$ concentrations. We also hope that our dataset can provide
auxiliary support for reanalysis datasets.
**5 $PM_{2.5}$ Variability from 1959 to 2022**
**5.1 Monthly $PM_{2.5}$ and Trend**
Figure 14 (a) shows the frequency of the estimated monthly $PM_{2.5}$ from 1959 to 2022, and Table 3
lists the maximum frequency for each region. The order of the concentrations with the greatest
frequency was Canada (8 $\mu g/m^3$), the United States (12 $\mu g/m^3$), Europe (18 $\mu g/m^3$), China (42 $\mu g/m^3$)
and India (64 $\mu g/m^3$). Canada and the United States are areas with less frequent $PM_{2.5}$ pollution.
$PM_{2.5}$ pollution occurs frequently in China and India. The results indicate that the $PM_{2.5}$
concentrations in developed countries are significantly lower than those in developing countries in
the Northern Hemisphere.
Figure 14 (b-f) shows the anomalies of the estimated monthly $PM_{2.5}$ concentration from 1959 to
2022, and Table 3 lists the trends for each region. The trends in each region from 1959 to 2022 are
all negative; however, the trend in India does not pass the significance test (p>0.05). The fastest
downward trend is in Europe, at -1.93 $\mu g/m^3$/decade. The trends in different regions vary at different
times. Positive trends are detected in the United States from 1959 to 1990, in Canada from 1959 to
1993, and in China and India from 1959 to 2012. The most rapid upward trend is observed in India,
at 3.35 $\mu g/m^3$/decade from 1959 to 2012. Negative trends are detected in the United States from
1991 to 2022, in Europe from 1959 to 1972 and from 1973 to 2022, and in China and India from
2013 to 2022. The most significant downward trend is observed in India, at -42.84 $\mu g/m^3$/decade.
These regional trends are similar to those of previous studies in different periods (Van Donkelaar et
al., 2010; Wang et al., 2012; Boys et al., 2014; Ma et al., 2016; Li et al., 2017; Hammer et al., 2020).
The trends in $PM_{2.5}$ concentration changes in different regions are closely associated with the
implementation of relevant policies. The earlier pollution control measures are taken, the earlier the
decreasing trend in the $PM_{2.5}$ concentration occurs, and the lower the threat of particulate matter
pollution is to humans. In 1997, the United States    EPA classified $PM_{2.5}$ as a hazardous substance
in the National Ambient Air Quality Standard, and subsequent regulations in 2006 further
strengthened the source control and management of fine particulate matter (Gilliam and Hall, 2016).
In 1988, the Canadian federal government enacted the Canadian Environmental Protection Act,
which enhanced the regulation of $PM_{2.5}$ (Davies, 1988). The European Union introduced the Air
Quality Directive in 1996, followed by multiple revisions and updates to regulate and restrict air



pollutants, including PM$_{2.5}$ (Kuklinska et al., 2015). However, Europe stands out due to its early
adoption of clean production practices in heavy industries since the 1970s. Since 2012, China has
implemented numerous regulations and standards for PM$_{2.5}$. For instance, the Monitoring Method
for Atmospheric Particulate Matter (PM$_{2.5}$) was issued in 2012, and the Chinese Ministry of
Environmental Protection released the Ambient Air Quality Standards in 2013, which include
emission standards for PM$_{2.5}$ (Zhao et al., 2016a). In 2009, the Indian Ministry of Environment and
Forests issued the National Ambient Air Quality Standards, which include control standards for air
pollutants, including PM$_{2.5}$. Since 2015, the Indian government launched the National Clean Air
Programme (NCAP) to improve air quality in India by implementing a series of measures to reduce
the emissions of PM$_{2.5}$ and other pollutants (Ganguly et al., 2020). These environmental regulations
have contributed significantly to the decline in PM$_{2.5}$ concentrations.

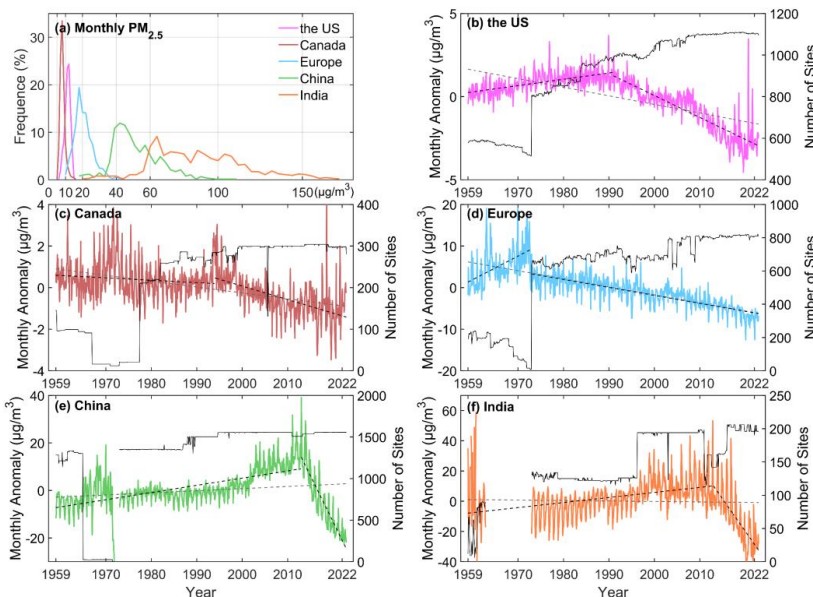

**Figure 14** Frequency (a) and anomalies (b-f) of monthly PM$_{2.5}$ from 1959 to 2022 in the United
States (the US), Canada, Europe, China, and India. The right Y-axis (b-f) is the monthly number of
sites.
**Table 3** The frequency and trend of the monthly PM$_{2.5}$ concentration from 1959 to 2022 in the
United States (the US), Canada, Europe, China and India.

| | Concentration Mode (µg/m³) and maximum frequency (%) | Trend (µg/m³/decade) | | |
|---|---|---|---|---|
| the US | 12 (24.3%) | -0.52* (1959-2022) | 0.38* (1959-1990) | -1.32* (1991-2022) |
| Canada | 8 (33.5%) | -0.28* (1959-2022) | -0.11* (1959-1993) | -6.48* (1994-2022) |
| Europe | 18 (19.4%) | -1.93 * (1959-2022) | 5.69 * (1959-1972) | -1.91* (1973-2022) |
| China | 42 (11.9%) | -0.89* | 3.04 * | -38.82* |



| | | (1959-2022) | (1959-2012) | (2013-2022) |
|---|---|---|---|---|
| *India* | 64 (9.1%) | -0.31<br>(1959-2022) | 3.35*<br>(1959-2012) | -42.84*<br>(2013-2022) |

The symbol * indicates passing the significance test, p<0.01; otherwise, not passing the significance test, p>0.05.

**5.2 Annual PM$_{2.5}$ and Distribution**

We analyze the spatial distribution of the multiyear average PM$_{2.5}$ concentration in each region, and we investigate the yearly variations in the spatial distribution based on the SDE and the average center, as shown in Figure 15. The mean center and SDE describe the periodic changes in the spatial distribution and dispersion of the PM$_{2.5}$ concentration in each region. The larger the ellipse area is, the more dispersed the spatial distribution of PM$_{2.5}$ is. The flatter the ellipse is, the stronger the spatial correlation of PM$_{2.5}$, and the direction of the major axis indicates the direction of the concentration.

The multiyear average PM$_{2.5}$ concentrations from 1959 to 2022 are 11.2 μg/m$^3$ in the United States, 8.2 μg/m$^3$ in Canada, 20.1 μg/m$^3$ in Europe, 51.3 μg/m in China, and 88.6 μg/m$^3$ in India. PM$_{2.5}$ concentrations in developed regions (North America and Europe) are significantly lower than those in developing regions (China and India).

For the United States, the concentration in the eastern region is greater than that in the western region. The PM$_{2.5}$ concentration at most sites in the eastern region is greater than 10 μg/m$^3$. Based on the area of the SDE, the spatial distribution is divided into three stages: 1959-1972, 1973-1976, and 1977-2022. The area decreases and then increases, indicating a changing trend in the spatial extent of the PM$_{2.5}$ concentration. The concentration distribution direction is east–west and rotates northward, and the mean center gradually moves northwest after 1977, indicating an increase in the PM$_{2.5}$ contribution in the western region.

For Canada, the concentrations in the eastern and western regions are greater than those in the central region. The area of the ellipse increases and then decreases. The concentration distribution direction is northwest-to-southeast, and the concentration rotates southward after 1977, indicating an increase in weight in the western region. The mean center gradually moves northwestward and then southeastward.

For Europe, high-concentration areas are mainly located in the central and eastern regions. The ellipse's area can be divided into three stages: 1959-1967, 1968-1972, and 1973-2022. The spatial variability decreases and then increases, corresponding to the mean centers moving north, south, and north. The concentration direction is northwest–southeast, and the major axis shortens after 1993, indicating that the directionality of the concentration weakens.

For China, high-concentration areas are in the central and eastern regions. The center of the SDE is located in the northeast region from 1965 to 1971, which may be related to Northeast China being the center of heavy industry during that period. After 1988, the area of the SDE increases significantly, and the center moves significantly southwestward and gradually northward after 2008. This finding indicates that the spatial distribution of PM$_{2.5}$ increases in a discrete pattern after 1988, and the concentration weight in the eastern region gradually increases. After 2008, the weight in the western region decreases again.





For India, the highest concentration is in the northern region, and the lowest concentration is in the
southern region. The area, shape, and mean center of the SDE show significant changes and can be
divided into three stages. The SDE flattens between 1959 and 1962. The flattening weakens, and
the area increases from 1963 to 1995. The spatial variability in $PM_{2.5}$ increases, and the mean center
moves southward. From 1996 to 2022, the flattening further weakens, the area decreases, the spatial
variability in $PM_{2.5}$ decreases, and the mean center shifts northward.
Above all, the concentration distributions in the United States and India exhibit an east–west pattern.
The concentration distribution in Canada and Europe shows a northwest-to-southeast concentration
gradient. In China, the $PM_{2.5}$ concentration distribution ranges from northeast to southwest. There
are strong correlations between the $PM_{2.5}$ concentration and the location of the sites in Europe and
Canada. However, the spatial correlation in India is gradually weakening, and the spatial dispersion
of $PM_{2.5}$ in China is increasing. Studies have shown that the variation in $PM_{2.5}$ based on the mean
center and the SDE is related to several factors, such as the energy structure, urbanization process,
population distribution and vegetation coverage (Shi et al., 2018; Wu et al., 2018; Li et al., 2019;
Wang et al., 2019; Lim et al., 2020; Qi et al., 2023).

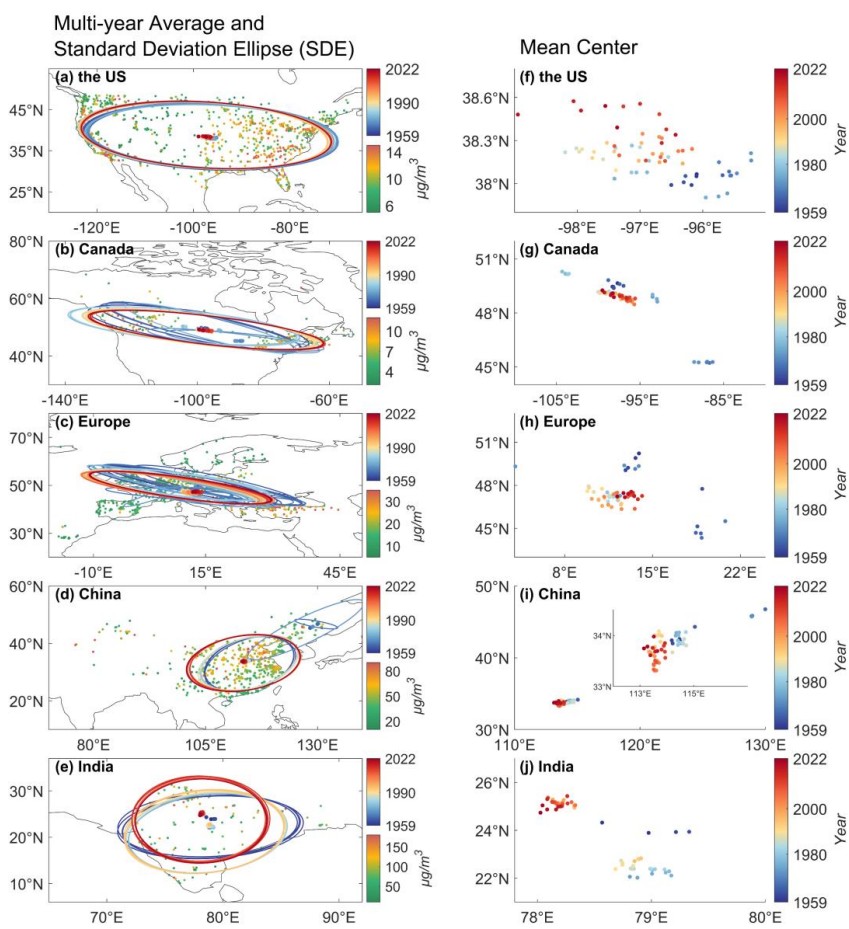


**Figure 15** The spatial distribution of the multiyear average and standard deviation ellipse (SDE) (a-e) and the mean center (f-j) of the PM$_{2.5}$ concentration from 1959 to 2022 in the United States (the US), Canada, Europe, China, and India. The mean center and SDE describe the changes in the spatial distribution. The larger the ellipse area is, the more dispersed the spatial distribution of PM$_{2.5}$ is. The flatter the ellipse is, the stronger the spatial correlation of PM$_{2.5}$ is. The direction of the major axis indicates the direction of the concentration.

**6 Conclusions**

This study uses a machine learning method to estimate daily PM$_{2.5}$ for 4011 terrestrial sites in the Northern Hemisphere from 1959 to 2022 based on hourly visibility and related meteorological variables. Eighty percent of the sample data are used to train the model, and 20% are used for testing. The model's performance and predictive ability are evaluated. We analyze the uncertainty and discuss the limitations of the dataset. We compare the estimated PM$_{2.5}$ with the PM$_{2.5}$ based on the satellite AOD and PM$_{2.5}$ of the reanalysis datasets. Finally, the PM$_{2.5}$ variability in each region over the past 64 years is analyzed. We hope our dataset will be useful for studying the atmospheric



environment, human health, and climate change and provide auxiliary support for assimilation.
Several key results of this study are described as follows:
**The most important variable** Visibility is the most important variable at 79.1% of the $PM_{2.5}$ sites,
as visibility can also be considered an indicator of $PM_{2.5}$ without fog or precipitation. Other
meteorological variables play a secondary role in the model, especially temperature and dew point
temperature. Visibility can serve as a good indicator of $PM_{2.5}$.
**Model performance and predictive ability.** The training results show that the slope between the
estimated $PM_{2.5}$ concentration and the monitored $PM_{2.5}$ concentration within the 95% confidence
interval is 0.946, the $R^2$ is 0.95, the RMSE is 7.0 $\mu g/m^3$, and the MAE is 3.1 $\mu g/m^3$. The test results
show that the slope between the predicted $PM_{2.5}$ concentration and the monitored $PM_{2.5}$
concentration is $0.862 \pm 0.0010$ within a 95% confidence interval, $R^2$ is 0.80, RMSE is 13.5 $\mu g/m^3$,
and MAE is 6.9 $\mu g/m^3$. The model shows good stability and predictive ability.
**Comparison with other datasets.** The estimated $PM_{2.5}$ concentration is consistent with the $PM_{2.5}$
concentration based on satellite AOD data at the monthly scale. The correlation coefficient of the
annual cycles in each region is greater than 0.96. Compared with the reanalysis data, there are some
differences among regions, which are closely related to the accuracy of emission inventories and
the vertical structures of aerosols.
**Monthly $PM_{2.5}$.** From 1959 to 2022, the $PM_{2.5}$ concentration at the highest frequency is 12 $\mu g/m^3$,
8 $\mu g/m^3$, 17 $\mu g/m^3$, 40 $\mu g/m^3$ and 63 $\mu g/m^3$, and the trends are -0.52 $\mu g/m^3$/decade, -0.28
$\mu g/m^3$/decade, -1.93 $\mu g/m^3$/decade, -0.89 $\mu g/m^3$/decade, and -0.31 $\mu g/m^3$/decade, respectively, for
the United States, Canada, Europe, China, and India. $PM_{2.5}$ concentrations in all regions show a
periodic increase and decrease from 1959 to 2022. The decreasing trends are -1.32 $\mu g/m^3$/decade
from 1991 to 2022 in the United States, -6.48 $\mu g/m^3$/decade from 1994 to 2022 in Canada, -1.91
$\mu g/m^3$/decade from 1973 to 2022 in Europe, and -38.82 $\mu g/m^3$/decade and -42.84 $\mu g/m^3$/decade from
2013 to 2022 in China and India, respectively. Although the $PM_{2.5}$ concentrations in developing
countries are significantly greater than those in developed countries, they have declined more
quickly in recent years.
**Annual $PM_{2.5}$.** The multiyear average $PM_{2.5}$ concentrations from 1959 to 2022 in the United States,
Canada, Europe, China, and India are 11.2 $\mu g/m^3$, 8.2 $\mu g/m^3$, 20.1 $\mu g/m^3$, 51.3 $\mu g/m^3$ and 88.6 $\mu g/m^3$,
respectively. Based on the features of the SDE and mean center, the spatial distribution of $PM_{2.5}$ has
more spatial variability in the United States, Canada, and Europe and less variability in China and
India. The changes in the mean center of the $PM_{2.5}$ concentration exhibit various patterns in each
region.
**7 Data Availability**
Daily $PM_{2.5}$ concentration data at 4011 sites in the Northern Hemisphere from 1959 to 2022 are
available at https://cstr.cn/18406.11.Atmos.tpdc.301127 (Hao et al., 2024).
**Competing Interests**
The contact author has declared that none of the authors has any competing interests.
**Acknowledgments**



This work was supported by the National Key Research & Development Program of China (2022YFF0801302) and the National Natural Science Foundation of China (41930970). The hourly visibility data are available at from https://mesonet.agron.iastate.edu/ASOS/. The hourly PM$_{2.5}$ data for the United States are available at https://www.epa.gov/aqs. The hourly PM$_{2.5}$ data for Canada are available at https://www.canada.ca. The hourly PM$_{2.5}$ data for Europe available at https://european-union.europa.eu. The hourly PM$_{2.5}$ data for China are available at https://www.cnemc.cn. The hourly PM$_{2.5}$ data for India are available at https://app.cpcbccr.com. The hourly visibility and meteorological data are available at https://www.weather.gov/asos. The monthly global PM$_{2.5}$ dataset for the Atmospheric Composition Analysis Group version V5.GL.04 (ACAG) are available at https://sites.wustl.edu/acag/datasets/surface-pm2-5/). The monthly PM$_{2.5}$ dataset of China High Air Pollutants (CHAP) are available at https://zenodo.org/records/6398971. The monthly PM$_{2.5}$ dataset of Modern-Era Retrospective Analysis for Research and Applications, version 2 (MERRA-2) are available at https://gmao.gsfc.nasa.gov. The monthly PM$_{2.5}$ of the Copernicus Atmosphere Monitoring Service (CAMS) reanalysis are available at https://ads.atmosphere.copernicus.eu/cdsapp#!/dataset/cams-global-reanalysis-eac4.

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
