# Peer review of "PM2.5 concentrations based on near-surface visibility at 4011 sites in the Northern 2 Hemisphere from 1959 to 2022"

_Earth System Science Data, 2024_

## Author Comment (AC1)

**PM$_{2.5}$ concentrations based on near-surface visibility in the Northern Hemisphere from 1959 to 2022**

We thank the referees for the constructive and helpful comments. We have carefully thought about the comments, made corresponding revisions to the manuscript and the datasets, and checked the manuscript carefully, which have substantially improved the manuscript and the datasets.

**Main modifications:**

■ Collected more PM$_{2.5}$ concentrations data (371 sites with more than 3-year observations) from openAQ in the Northern Hemisphere in Section 2.2.6, increasing the coverage in the NH.

■ Used visibility data from ISD instead of the original visibility data in Section 2.3, which resulting in more than 1000 stations added than previous version. Based on ISD visibility, the distances decrease significantly. And the upper limit is set to 100 km.

■ Added the comparisons on the daily/monthly scale and before/after 2010 in Section 4.1, to evaluate the predictive ability of the model and the consistency of estimated PM$_{2.5}$ concentration.

■ Used GAMM to analyze the interannual trends and spatial patterns on the regional scale due to irregular site distribution in Section 5.

■ Adjusted the structure and content of the manuscript. And all figures and tables have been modified or replaced.

**Response to Anonymous Referee #1**

*Hao et al. used the visibility to estimate the historical PM$_{2.5}$ concentration in the northern hemisphere in the past 60 years. Overall, the topic is very interesting and the manuscript is well-organized. However, the manuscript still suffers from some major flaws and thus I recommend the manuscript for publication on ESSD after the following comments have been well addressed.*

**Comment 1.1**. *Visibility is a useful tool to estimate the long-term PM$_{2.5}$ concentration during a long period. However, the accuracy based on visibility was generally less than that based on AOD. Why do not you use the combination scheme of AOD and visibility? For instance, you could use AOD during 2000-2022, and use visibility before 2000. I think you should evaluate the performances of two schemes and compared the difference in your study.*

● **Response 1.1:**

Near-surface visibility quantifies surface optical concentration of aerosols, which is directly related to the surface mass concentration, i.e., PM$_{2.5}$. AOD describes the column total optical concentration of atmospheric aerosols, which indirectly correlates with PM$_{2.5}$ bridging by the atmospheric aerosol scale height. These differences are discussed in Section 4.3. Independent evaluation in this study shows PM$_{2.5}$ concentration based on visibility is reliable with high correlation coefficients and low root mean square errors. More important, the visibility derived PM$_{2.5}$ concentration is long-term and consistent and can provide time series from 1959 to 2022. However,

satellite AOD based methods can provide time series of $PM_{2.5}$ since 2000. To avoid inconsistency, we would like to keep same input data.

***Comment 1.2***. *Visibility station is scattered around the world. Why do you only focus on China, Europe, US, and India? I think the estimates of long-term $PM_{2.5}$ concentrations across the northern hemisphere might be more valuable. You could even construct the full-coverage grid-based $PM_{2.5}$ dataset across the northern hemisphere.*

● **Response 1.2:**

Thank you for your suggestion.

(1) We have further collected more $PM_{2.5}$ observations and used visibility data from ISD to increase the coverage. $PM_{2.5}$ concentrations of 1012 sites are added, as shown in Figure 1.

(2) We are aiming at establishing a long-term site-scale dataset in this study. We are trying our best to build a grid-scale $PM_{2.5}$ product based on visibility by another method. Therefore, this study does not involve grid products.

***Comment 1.3***. *Section 3.2.2: The validation of constructed $PM_{2.5}$ dataset in recent years might be not enough because the major novelty of this study is a long-term estimate. Thus, the authors should add more examinations of $PM_{2.5}$ estimates before 2010 especially in China and India. I think the authors could search many previous references to obtain these ground-level observations.*

● **Response 1.3:**

We have added the examinations of $PM_{2.5}$ concentration before/after 2010 in Section 4.

***Comment 1.4***. *I think the comparison of your dataset with other reanalysis data might be not very necessary because the dataset in this study is site-based instead of grid-based. I think you must confirm your dataset is much superior to all of the previous reanalysis dataset if you want to compare them*

● **Response 1.4:**

We have removed the comparisons with the reanalysis data.

***Comment 1.5***. *Figure 14: Why do the $PM_{2.5}$ in India experience dramatic decreases from 2010 to 2022? I think India proposed clean air policy since 2019. The authors should test the observations to examine whether the estimate is right.*

● **Response 1.5:**

We have checked the estimated $PM_{2.5}$ concentrations and investigated some studies about the trends in India. Our results are similar to previous studies.

For example, Singh et al. (2021) has found that $PM_{2.5}$ concentration of five major cities in India show a downward trend from 2014 to 2019, and the largest declining trend (-4.2 $\mu g/m^3$ per year) is in New Delhi. Ravindra et al. (2024) also finds that the trend in New Delhi is about -5 $\mu g/m^3$ per year from 2014 to 2020.

---

## Author Comment (AC2)

**PM$_{2.5}$ concentrations based on near-surface visibility in the Northern Hemisphere from 1959 to 2022**

We thank the referees for the constructive and helpful comments. We have carefully thought about the comments, made corresponding revisions to the manuscript and the datasets, and checked the manuscript carefully, which have substantially improved the manuscript and the datasets.

**Main modifications:**

- Collected more PM$_{2.5}$ concentrations data (371 sites with more than 3-year observations) from openAQ in the Northern Hemisphere in Section 2.2.6, increasing the coverage in the NH.

- Used visibility data from ISD instead of the original visibility data in Section 2.3, which resulting in more than 1000 stations added than previous version. Based on ISD visibility, the distances decrease significantly. And the upper limit is set to 100 km.

- Added the comparisons on the daily/monthly scale and before/after 2010 in Section 4.1, to evaluate the predictive ability of the model and the consistency of estimated PM$_{2.5}$ concentration.

- Used GAMM to analyze the interannual trends and spatial patterns on the regional scale due to irregular site distribution in Section 5.

- Adjusted the structure and content of the manuscript. And all figures and tables have been modified or replaced.

**Response to Anonymous Referee #2**

*Hao et al. utilized a machine learning method to estimate a long-term global PM$_{2.5}$ dataset based on visibility data at a site scale. Comprehensive validation and analysis have confirmed the reliability and value of this dataset. However, there are some major issues that must be addressed before considering the manuscript for publication. The specific comments are as follows.*

***Comment 2.1***. *L23-31: The representativeness of spatially distributed sparse station monitoring data for average concentrations on a national scale needs careful consideration. In China, PM$_{2.5}$ monitoring stations are predominantly located in urban areas, where concentrations tend to be higher than in rural areas. Additionally, the methodology for calculating trends warrants clarification. Calculating regional trends across these locations is challenging due to the uneven distribution of monitoring sites. Chang et al. (2017) noted that the European network is more sparsely populated across its northern and eastern regions and therefore a simple average of the individual trends at each site does not yield an accurate regional trend. More robust conclusions could be drawn when estimating the spatiotemporal full-coverage dataset. Reference: Kai-Lan Chang, Irina Petropavlovskikh, Owen R. Cooper, Martin G. Schultz, Tao Wang; Regional trend analysis of surface ozone observations from monitoring networks in eastern North America, Europe and East Asia. Elementa: Science of the Anthropocene 1 January 2017; 5 50. doi: https://doi.org/10.1525/elementa.243*

- **Response 2.1:**

Thank you for your suggestion. We have used GAMM (Chang et al., 2017) to analyze the regional trends and spatial patterns in Section 5.

*Comment 2.2. L39-141: The content is repeated in the caption of Figure 1.*

- **Response 2.2:**

Thank you for your correction. We have made modifications.

*Comment 2.3. L197: Does "2000" in sites as of 2000 refer to 2022 or 2020? Figure 1 indicates the sites in China have existed for only about ten years.*

- **Response 2.3:**

Thank you for your correction. We have made modifications.

*Comment 2.4. L332: Please provide the full name of the abbreviation "CART".*

- **Response 2.4:**

Thank you for your correction. We have made modifications.

*Comment 2.5. How are $PM_{2.5}$, visibility and meteorological data matched spatially, and what is the distance between $PM_{2.5}$ and meteorological monitoring stations? Are there multiple $PM_{2.5}$ sites that match the same meteorological and visibility stations, thereby providing the same features and different labels for the samples of these sites? This scenario is counterfactual.*

- **Response 2.5:**

We have added details on the spatiotemporal matching between visibility station and $PM_{2.5}$ site in Section 2.4.

*Comment 2.6. The verification method for the machine learning model may not be convincing, even if the cross-validation based on samples was used. Given the study aims to establish a long-term $PM_{2.5}$ dataset, especially for historical periods lacking surface monitoring, the temporal generalization performance of the model is crucial. It is necessary to evaluate the performance based on data from the period not included in the training dataset. For instance, the model could be trained on data from before 2020 and tested on data from after 2020.*

- **Response 2.6:**

We sort the sample data by time, the first 80% of sample data is the training set, and the last 20% is the test set, which has been stated in section 2.6.

*Comment 2.7. L615: "Elevation of Meteorological Station" should be corrected to "Elevation of Visibility Station" in Figure 9. The same problem occurs in Figure 10.*

- **Response 2.7:**

Thank you for your correction. We have made modifications.

*Comment 2.8. L805: There is no section 2.6.3, please check the full text.*

- **Response 2.8:**

Thank you for your correction. We have made modifications.

---

## Author Comment (AC3)

**PM$_{2.5}$ concentrations based on near-surface visibility in the Northern Hemisphere from 1959 to 2022**

We thank the referees for the constructive and helpful comments. We have carefully thought about the comments, made corresponding revisions to the manuscript and the datasets, and checked the manuscript carefully, which have substantially improved the manuscript and the datasets.

**Main modifications:**

■ Collected more PM$_{2.5}$ concentrations data (371 sites with more than 3-year observations) from openAQ in the Northern Hemisphere in Section 2.2.6, increasing the coverage in the NH.

■ Used visibility data from ISD instead of the original visibility data in Section 2.3, which resulting in more than 1000 stations added than previous version. Based on ISD visibility, the distances decrease significantly. And the upper limit is set to 100 km.

■ Added the comparisons on the daily/monthly scale and before/after 2010 in Section 4.1, to evaluate the predictive ability of the model and the consistency of estimated PM$_{2.5}$ concentration.

■ Used GAMM to analyze the interannual trends and spatial patterns on the regional scale due to irregular site distribution in Section 5.

■ Adjusted the structure and content of the manuscript. And all figures and tables have been modified or replaced.

**Response to Anonymous Referee #3**

*It is a very interesting paper that estimates the long-term PM$_{2.5}$ concentration in the northern hemisphere using machine learning, and I believe this dataset is meaningful. Still, there are several questions I want to ask.*

*Comment 3.1. Consider presenting the methodology in a tabular format, summarizing key details such as the number of sites, time span, time resolution, and other pertinent information across different regions. It may be clearer.*

● **Response 3.1:**

Thank you for your suggestion. We have added a table to describe the data information in Table 1.

*Comment 3.2. Is there variation in regional rankings for variable importance?*

● **Response 3.2:**

We have added the most important variables of different regions in Figure 2 in Section 3.1.

*Comment 3.3. The trends in the Indian seems to be very different. Is this attributed to the overestimation of PM$_{2.5}$ in dusty areas, as mentioned in Line 793? Furthermore, Figure 12 indicates ACAG has superior agreement with your dataset compared to MERRA-2 and CAMS. Could this disparity be attributed to differences in resolution or time span? In addition, given the high time resolution characteristic of your dataset, a daily comparison would be interesting.*

- **Response 3.3:**

(1) We have checked the estimated $PM_{2.5}$ concentration and investigated some studies about the trend in India. For example, Singh et al. (2021) have found that $PM_{2.5}$ concentration of five major cities in India show a downward trend from 2014 to 2019, and the largest declining trend (-4.2 μg/m$^3$ per year) is in New Delhi. Ravindra et al. (2024) also finds that the trend in New Delhi is about -5 μg/m$^3$ per year from 2014 to 2020.

(2) We have added the comparisons on the daily scale in Section 4.1.

*Comment 3.4. The authors should clarify how $PM_{2.5}$ and visibility data were matched spatially and temporally. It would be good to clarify whether the machine learning is based on hourly data or corresponding daily mean data.*

- **Response 3.4:**

We have added the spatiotemporal matching method and clarified that the machine learning is based on daily mean data in Section 2.4.

*Comment 3.5. If $PM_{2.5}$ concentration and visibility data have differing daily hour intervals, how were they aligned? Furthermore, is it really reasonable to include stations located several hundred kilometers apart in the training dataset?*

- **Response 3.5:**

We have added the temporal matching method in section 2.4. We have used the visibility data from ISD instead of original visibility data and the upper limit of distance is set to 100 km.

*Additionally, the authors may attention to details, such as:*

*Comment 3.6. In Line 940, it would be preferable to maintain consistency by unifying "20%" and "80%".*

- **Response 3.6:**

Thank you for your correction. We have made modifications.

*Comment 3.7. Line 947 requires a ".".*

- **Response 3.7:**

Thank you for your correction. We have made modifications.

*Comment 3.8. Line 950 should specify "$PM_{2.5}$ concentration" rather than "$PM_{2.5}$" (this problem exists in the whole paper). And maybe discussions on past decade trends should be moved to the annual section for better organization.*

- **Response 3.8:**

Thank you for your suggestion. We have made modifications and adjusted the structure.

---

## Author Response (AR2)

**PM$_{2.5}$ concentrations based on near-surface visibility in the Northern Hemisphere from 1959 to 2022**

We thank the referees for the comment and carefully thought about the comment.

**Response to Anonymous Referee #2**

*Comment. I am curious why the test results sorted by time (in the reviewed version) are only slightly lower than the random test results (in the first submission) (0.80 to 0.79), and even the test results on a site scale are better in China and India. Typically, test results divided by time are significantly lower than those divided randomly. For example, Yang et al. (2022) developed a model to estimate PM2.5 concentrations and showed that sample-based cross-validation (CV) and date-based CV yielded R2 of 0.92 and 0.63, respectively. (Reference: "Geographical and temporal encoding for improving the estimation of PM2.5 concentrations in China using end-to-end gradient boosting," Remote Sensing of Environment, 2022).*

● **Response:**

(1) This study is based on the time series of the site for modeling and prediction (not based on spatial distribution prediction), which will capture the historical variation easily at site scale, avoiding the influence of similar environments (the similar values of variables) in different location due to the strong spatial variability of PM$_{2.5}$ concentration.

(2) The dependence of variables shows that visibility is the most important variable, indicating that it is an indicator of PM$_{2.5}$ concentration. Especially in China, the contribution is even greater than 90%. In addition, we also discussed the differences in visibility-based and AOD-based PM$_{2.5}$ concentrations in section 4.3.

(3) **Compared to the previous version, the main reason for the better performance in China and India is that more visibility stations were added, greatly reducing the errors caused by spatial distance between visibility station and PM$_{2.5}$ site.**